# Contraction-induced endocardial *id2b* plays a dual role in regulating myocardial contractility and valve formation

**Shuo Chen[1,2,3†], Jinxiu Liang[1,2,3†], Jie Yin[1,2,3], Weijia Zhang[1,2,3], Peijun Jiang[1,2,3], Wenyuan Wang[4,5], Xiaoying Chen[6,7], Yuanhong Zhou[8], Peng Xia[8], Fan Yang[6,7], Ying Gu[1,2,3], Ruilin Zhang[4,5], Peidong Han[1,2,3]***

[1]Department of Cardiology, Center for Genetic Medicine, The Fourth Affiliated Hospital of School of Medicine, and International School of Medicine, International Institutes of Medicine, Zhejiang University, Yiwu, China; [2]Division of Medical Genetics and Genomics, The Children's Hospital, Zhejiang University School of Medicine, National Clinical Research Center for Child Health, Hangzhou, China; [3]Institute of Genetics, Zhejiang University School of Medicine, Hangzhou, China; [4]TaiKang Medical School (School of Basic Medical Sciences), Wuhan University, Wuhan, China; [5]Hubei Provincial Key Laboratory of Developmentally Originated Disease, Wuhan, China; [6]Department of Biophysics, and Kidney Disease Center of the First Affiliated Hospital, Zhejiang University School of Medicine, Hangzhou, China; [7]Liangzhu Laboratory, Zhejiang University Medical Center, Hangzhou, China; [8]Zhejiang Provincial Key Laboratory for Cancer Molecular Cell Biology, Life Sciences Institute, Zhejiang University, Hangzhou, China

**\*For correspondence:**
hanpd@zju.edu.cn

†These authors contributed equally to this work

**Competing interest:** The authors declare that no competing interests exist.

## eLife Assessment

This study presents a **valuable** finding that the biomechanical force of heart contractility is required for robust endocardial id2b expression, which in return promotes valve development and myocardial function through upregulation of Neuregulin 1. The data were collected and analyzed using **solid** methodology and can be used as a starting point for deeper mechanistic insights into the genetic programs regulating endocardial-myocardial crosstalk during heart development.

**Abstract** Biomechanical cues play an essential role in sculpting organ formation. Comprehending how cardiac cells perceive and respond to biomechanical forces is a biological process with significant medical implications that remains poorly understood. Here, we show that biomechanical forces activate endocardial *id2b* (inhibitor of DNA-binding 2b) expression, thereby promoting cardiac contractility and valve formation in zebrafish. Taking advantage of the unique strengths of zebrafish, particularly the viability of embryos lacking heartbeats, we systematically compared the transcriptomes of hearts with impaired contractility to those of control hearts. This comparison identified *id2b* as a gene sensitive to blood flow. By generating a knock-in reporter line, our results unveiled the presence of *id2b* in the endocardium, and its expression is sensitive to both pharmacological and genetic perturbations of contraction. Furthermore, *id2b* loss-of-function resulted in progressive heart malformation and early lethality. Combining RNA-seq analysis, electrophysiology, calcium imaging, and echocardiography, we discovered profound impairment in atrioventricular (AV) valve formation and defective excitation-contraction coupling in *id2b* mutants. Mechanistically, deletion of *id2b* reduced AV endocardial cell proliferation and led to a progressive increase in retrograde blood flow. In the myocardium, *id2b* directly interacted with the bHLH component *tcf3b*

(transcription factor 3b) to restrict its activity. Inactivating *id2b* unleashed its inhibition on *tcf3b*, resulting in enhanced repressor activity of *tcf3b*, which subsequently suppressed the expression of *nrg1* (neuregulin 1), an essential mitogen for heart development. Overall, our findings identify *id2b* as an endocardial cell-specific, biomechanical signaling-sensitive gene, which mediates intercellular communications between endocardium and myocardium to sculpt heart morphogenesis and function.

## Introduction

The heart develops with continuous contraction, and biomechanical cues play an essential role in cardiac morphogenesis (*Duchemin et al., 2019*; *Sidhwani and Yelon, 2019*). Blood flow is directly sensed by the surrounding endocardium, which undergoes multiscale remodeling during zebrafish heart development. In the atrioventricular canal (AVC) endocardium, oscillatory flow promotes valvulogenesis through transient receptor potential (TRP) channel-mediated expression of Krüppel-like factor 2a (*klf2a*) (*Vermot et al., 2009*; *Heckel et al., 2015*; *Gálvez-Santisteban et al., 2019*). Meanwhile, mechanical forces trigger ATP-dependent activation of purinergic receptors, inducing expression of nuclear factor of activated T cells 1 (*nfatc1*) and subsequent valve formation (*Fukui et al., 2021*). In the chamber endocardium, blood flow induces endocardial cells to adopt chamber- and region-specific cell morphology during cardiac ballooning (*Dietrich et al., 2014*). A recent study further emphasized that blood flow is essential for endocardial cell accrual in assembling the outflow tract (OFT) (*Sidhwani et al., 2020*). Beyond their role in endocardial cells, proper biomechanical cues are indispensable for shaping the myocardium. For instance, in contraction-compromised *tnnt2a* (*Staudt et al., 2014*) and *myh6* (*Peshkovsky et al., 2011*) mutants, trabeculation is markedly reduced. Moreover, apart from the tissue-scale regulatory effect, the shape changes (*Auman et al., 2007*) and myofibril content (*Lin et al., 2012*) at the single-cardiomyocyte level are also sculpted by the interplay of contractility and blood flow in the developing heart.

In ventricular myocardium morphogenesis, biomechanical forces coordinate intra-organ communication between endocardial and myocardial cells by regulating bone morphogenetic protein (BMP), Nrg/Erbb, and Notch signaling. The Nrg-Erbb axis stands as one of the most extensively studied signaling pathways mediating cell-cell communications in the heart (*Gassmann et al., 1995*; *Lee et al., 1995*; *Meyer and Birchmeier, 1995*; *Zhang et al., 2021*). In particular, endocardial Notch activity induced by cardiac contraction promotes the expression of Nrg, which then secretes into the extracellular space, binding to Erbb2/4 receptor tyrosine kinases on cardiomyocyte and promoting their delamination (*Lai et al., 2010*; *Rentschler et al., 2002*; *Liang et al., 2025*). In agreement with the pivotal role of this signaling pathway in heart development, genetic mutations in zebrafish *nrg2a* and *erbb2* result in severely compromised trabeculae formation (*Liu et al., 2010*; *Rasouli and Stainier, 2017*; *Han et al., 2016*).

The development of cardiomyocytes encompasses the specification of subcellular structure, metabolic state, gene expression profile, and functionality (*Guo and Pu, 2020*; *Alvarez-Dominguez and Melton, 2022*). The rhythmic contraction of cardiomyocytes relies on precisely regulated excitation-contraction coupling (E-C coupling), transducing electrical activity into contractile forces. This intricate signaling cascade involves membrane action potential, calcium signaling, and sarcomeric structure (*Bers, 2002*; *Xu et al., 2024*). Specifically, membrane depolarization triggers the opening of L-type calcium channel (LTCC), allowing calcium influx. The calcium signaling then activates the ryanodine receptor on the sarcoplasmic reticulum membrane, releasing additional calcium (*Bers, 2002*). E-C coupling is essential for heart development, as evidenced by the complete silence of the ventricle and reduced cardiomyocyte number in *cacna1c* (LTCC α1 subunit in zebrafish) mutant (*Rottbauer et al., 2001*). Beyond its role in modulating cardiac structure formation, previous studies indicate that Nrg-Erbb2 signaling is also necessary for cardiac function, as *erbb2* mutants exhibit severely compromised fractional shortening and an immature conduction pattern (*Liu et al., 2010*; *Samsa et al., 2015*).

Id (inhibitor of DNA-binding) proteins belong to the helix-loop-helix (HLH) family and function as transcriptional repressors (*Benezra et al., 1990*). Notably, Id2 lacks a DNA-binding domain and forms a heterodimer with other bHLH proteins, acting in a dominant-negative manner (*Wong et al., 2012*). Id2 plays a crucial role in heart development, and its genetic deletion results in severe cardiac defects in mice (*Jongbloed et al., 2011*; *Moskowitz et al., 2007*; *Moskowitz et al., 2011*). In zebrafish, *id2a*

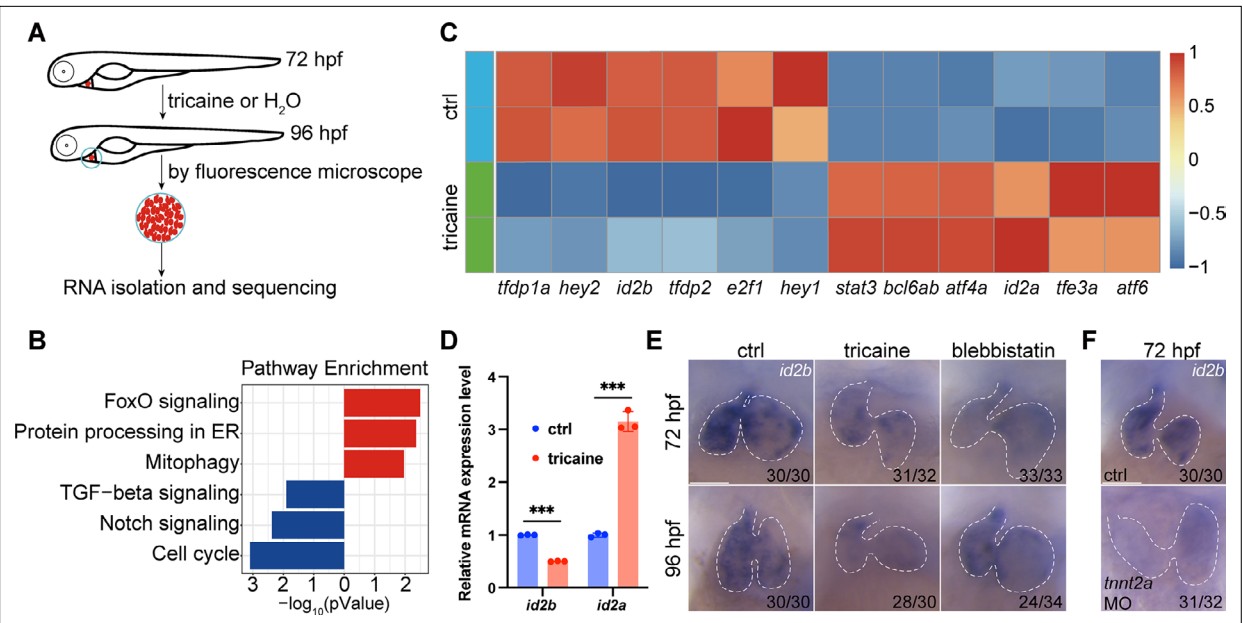

**Figure 1.** Identification of *id2b* as a blood flow-sensitive gene. (**A**) Schematic showing the experimental procedures, including treatment, heart collection, and RNA-sequencing of zebrafish embryonic hearts. (**B**) KEGG enrichment analysis depicting differentially expressed genes encoding transcription factors and transcriptional regulators between control (ctrl) and tricaine-treated embryonic hearts. Red and blue rectangles represent upregulated and downregulated gene sets, respectively. |log$_2$fold change|≥0.585, adjusted p-value<0.1. Each replicate contains approximately 1000 hearts. (**C**) Heatmap exhibiting representative genes from KEGG pathways mentioned in (**B**). (**D**) Quantitative real-time PCR (qRT-PCR) analysis of *id2b* and *id2a* mRNA in control and tricaine-treated embryonic hearts. Data were normalized to the expression of *actb1*. Each sample contains ~1000 embryonic hearts. N=3 biological replicates. (**E**) In situ hybridization of *id2b* in 72 hr post-fertilization (hpf) and 96 hpf ctrl, tricaine (1 mg/mL), and blebbistatin (10 µM)-treated embryos. Numbers at the bottom of each panel indicate the ratio of representative staining. (**F**) In situ hybridization showing reduced *id2b* expression in *tnnt2a* morpholino-injected embryos at 72 hpf compared to control. Data are presented as mean ± s.e.m. Unpaired two-tailed Student's t-tests were used to determine statistical significance. ***p<0.001. Scale bars, 50 µm.

and *id2b* are homologs of the mammalian *Id2* gene. However, their expression pattern and function in the zebrafish heart remain largely unknown. In the present study, we identified that *id2b* is specifically expressed in endocardial cells of the developing heart, and its expression is regulated by cardiac contraction and blood flow. Genetic deletion of *id2b* led to impaired AV valve formation and reduced cardiac contractility. Therefore, *id2b* serves as a crucial mediator linking biomechanical cues to heart morphogenesis.

## Results

### Transcriptome analysis identifies *id2b* as a blood flow-sensitive gene

Blood circulation is dispensable for early embryonic development in zebrafish, presenting an ideal model to investigate biomechanical cues influencing heart morphogenesis. To identify genes affected by cardiac contraction or blood flow, we treated *myl7:mCherry* zebrafish embryos with tricaine to inhibit cardiac contractility from 72 hr post-fertilization (hpf) to 96 hpf. Hearts from control and tricaine-treated zebrafish embryos were manually collected under a fluorescence stereoscope as previously reported (*Burns and MacRae, 2006*). Subsequently, approximately 1000 hearts from each group were subjected to RNA-seq (*Figure 1A*). A total of 4530 genes with differential expression were identified, comprising 2013 upregulated and 2517 downregulated genes. With a specific focus on identifying key transcription factors (TFs) affected by perturbing biomechanical forces, differentially expressed genes (DEGs) encoding TFs were enriched into signaling pathways through KEGG analysis. Interestingly, our analysis identified several pathways known to be involved in heart development, including the transforming growth factor beta signaling and Notch signaling pathways (*Figure 1B*). In particular, the scaled expression levels of the top 6 DEGs (|log$_2$FC|≥0.585), exhibiting up- or downregulation, were listed (*Figure 1C*). Intriguingly, *Id2* has been shown to regulate murine AV valve formation, a

process notably influenced by alterations in blood flow directionality. Moreover, loss of *Id2* leads to malformation of both the arterial and venous poles of the heart and disrupts AV valve morphogenesis (*Jongbloed et al., 2011*; *Moskowitz et al., 2011*). Therefore, we interrogated the expression and function of *id2b* in developing embryos.

Quantitative real-time PCR (qRT-PCR) analysis of purified embryonic hearts revealed a significant reduction in *id2b* mRNA levels and an increase in *id2a* levels following tricaine treatment from 72 to 96 hpf compared to controls (*Figure 1D*). Furthermore, in situ hybridization was performed to visualize *id2b* expression under tricaine or 10 µM blebbistatin (an inhibitor of sarcomeric function and cardiac contractility) treatment from 48 to 72 or from 72 to 96 hpf as previously described (*Gálvez-Santisteban et al., 2019*). Consistently, our results showed a reduction in *id2b* signal in contraction-deficient hearts compared to the control (*Figure 1E*). In cardiomyocytes, *tnnt2a* encodes a key sarcomeric protein essential for contractility. Similarly, injection of a previously characterized *tnnt2a* morpholino (*Sehnert et al., 2002*) at the one-cell stage also led to compromised contraction and diminished expression

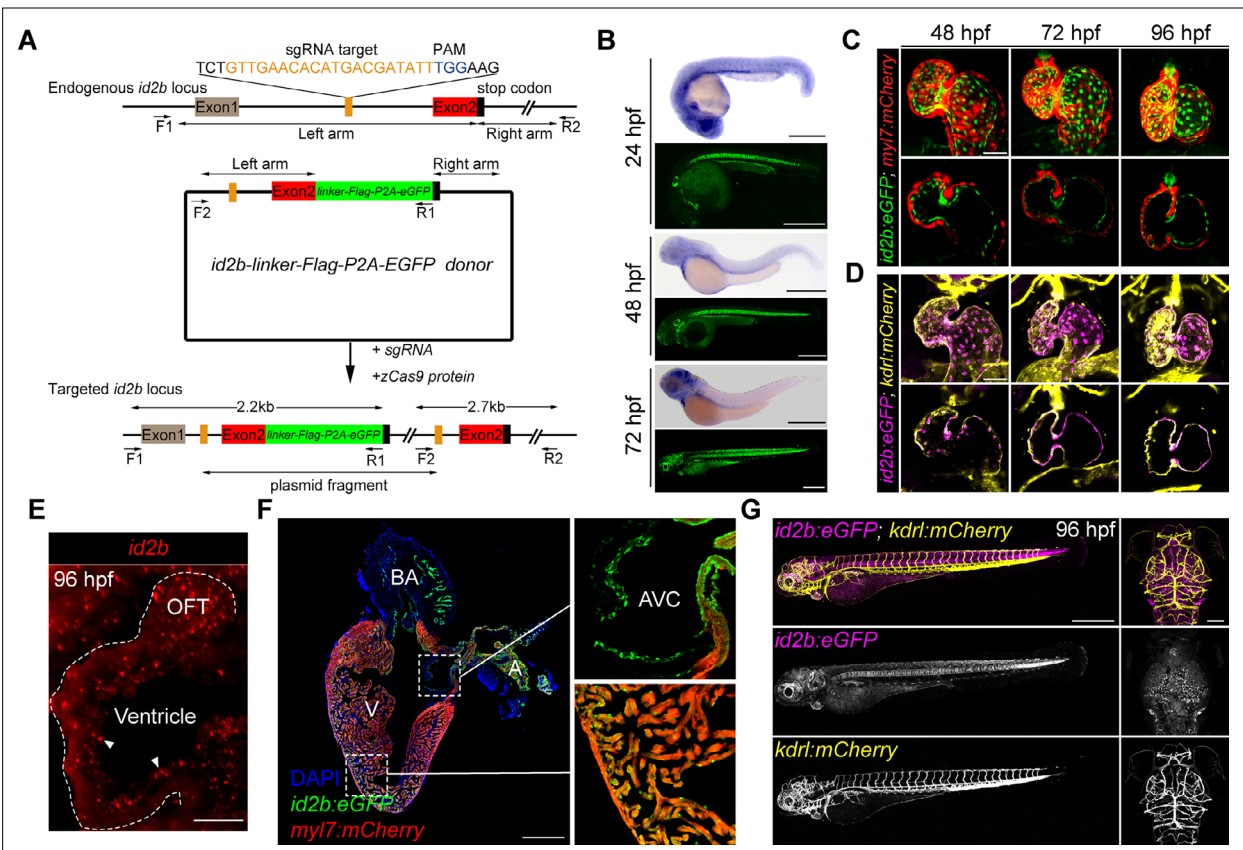

**Figure 2.** The spatiotemporal expression of *id2b*. (**A**) Schematic of the intron targeting-mediated *eGFP* knock-in at the *id2b* locus using the CRISPR-Cas9 system. The sgRNA targeting sequence and the protospacer adjacent motif (PAM) sequence are shown in orange and blue, respectively. The donor plasmid comprises left and right arm sequences and a *linker-FLAG-P2A-eGFP* cassette denoted by black lines with double arrows and green box, respectively. The *linker-FLAG-P2A-eGFP* cassette was integrated into the *id2b* locus upon co-injection of the donor plasmid with sgRNA and zCas9 protein, enabling detection by PCR using two pairs of primers (F1, R1 and F2, R2) - the former length yielding a length of about 2.2 kb and the latter about 2.7 kb. (**B**) Zebrafish *id2b* expression pattern, as indicated by in situ hybridization of embryos at designated time points, was consistent with the fluorescence localization of *id2b:eGFP*, revealing expression in the heart, brain, retina, notochord, pronephric duct, and other tissues. (**C**) Maximum intensity projections (top) and confocal sections (bottom) of *id2b:eGFP; Tg(myl7:mCherry)* hearts at designated time points. (**D**) Maximum intensity projections (top) and confocal sections (bottom) of *id2b:eGFP; Tg(kdrl:mCherry)* embryos at specific time points. Magenta, *id2b:eGFP*; yellow, *kdrl:mCherry*. (**E**) RNAscope analysis of *id2b* in 96 hr post-fertilization (hpf) embryonic heart. White dashed line outlines the heart. OFT, outflow tract. (**F**) Immunofluorescence of adult *id2b:eGFP; Tg(myl7:mCherry)* heart section (left panel). Enlarged views of boxed areas are shown in the right panel. Green, eGFP; red, mCherry; blue, DAPI. BA, bulbus arteriosus; V, ventricle; A, atrium; AVC, atrioventricular canal. (**G**) Confocal z-stack maximum intensity projection of *id2b:eGFP;Tg(kdrl:mCherry)* embryos at 96 hpf showing the whole body (lateral view) and the head (top view). Scale bars, 500 µm (**B**, **F**, left, and **G**), 50 µm (**C and D**), 25 µm (**E**), 100 µm (**G**, right).

of *id2b* (*Figure 1F*). Taken together, these results indicate that biomechanical cues are essential for activating *id2b* in embryonic hearts.

## Visualization of the spatiotemporal expression of *id2b* in developing embryos

Due to technical challenges in visualizing the cell-type-specific expression of *id2b* in the developing heart using whole-mount in situ hybridization, we employed an intron targeting-mediated approach (*Li et al., 2015*) to generate a knock-in *id2b:eGFP* reporter line. This method allowed us to achieve specific labeling without perturbing the integrity and function of the endogenous gene (*Li et al., 2015*; *Figure 2A*). Comparison of *id2b:eGFP* fluorescence with in situ hybridization at 24, 48, and 72 hpf revealed that the reporter signal closely recapitulates the endogenous *id2b* expression pattern. The fluorescence was notably enriched in the heart, brain, retina, and notochord (*Figure 2B*), mirroring observations from a previously reported *id2b* transgenic line generated through BAC-mediated recombination (*Förster et al., 2017*).

To further elucidate the spatiotemporal expression of *id2b* in developing hearts, we crossed *id2b:eGFP* with *myl7:mCherry* or *kdrl:mCherry*, labeling cardiomyocytes or endocardial cells,

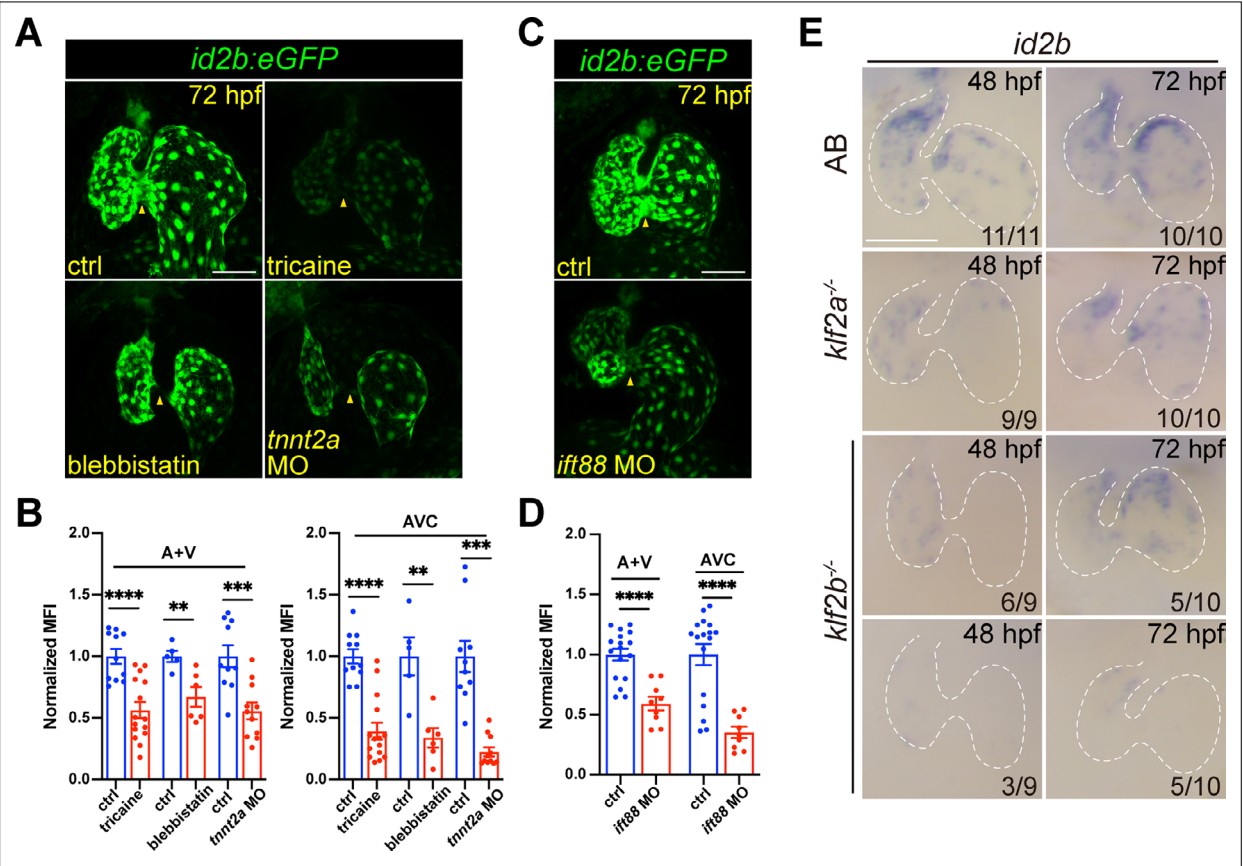

**Figure 3.** Cardiac contraction promotes endocardial *id2b* expression through primary cilia. (**A**) Representative confocal z-stack (maximal intensity projection) of *id2b:eGFP* embryos under different conditions: control, tricaine-treated, blebbistatin-treated, and *tnnt2a* morpholino-injected. Images were captured using the same magnification. (**B**) Quantification of mean fluorescence intensity (MFI) of *id2b:eGFP* in the working myocardium (atrium and ventricle, A+V) and atrioventricular canal (AVC) in (**A**). Data normalized to the MFI of control hearts. n=(11, 15) (ctrl versus tricaine); n=(5, 6) (ctrl versus blebbistatin); n=(10,11) (ctrl versus *tnnt2a* MO). (**C**) Representative confocal z-stack (maximal intensity projection) of control and *ift88* morpholino-injected *id2b:eGFP* embryos. (**D**) Normalized MFI of *id2b:eGFP* in the working myocardium (A+V) and AVC in (**C**). n=(17, 9). (**E**) Whole-mount in situ hybridization showing *id2b* expression in control, *klf2a$^{-/-}$*, and *klf2b$^{-/-}$* embryos at 48 hr post-fertilization (hpf) and 72 hpf. Numbers at the bottom of each panel indicate the ratio of representative staining. Data are presented as mean ± s.e.m. Unpaired two-tailed Student's t-tests were used to determine statistical significance. **p<0.01, ***p<0.001, ****p<0.0001. Scale bars, 50 µm.

The online version of this article includes the following figure supplement(s) for figure 3:

**Figure supplement 1.** Blood flow and bone morphogenetic protein (BMP) signaling independently activates *id2b* expression.

respectively. Confocal images revealed minimal, if any, presence of *id2b:eGFP* in *myl7:mCherry⁺* cardiomyocytes (*Figure 2C*). In sharp contrast, clear co-localization between *id2b:eGFP* and *kdrl:mCherry* was evident at 48, 72, and 96 hpf (*Figure 2D*). Endocardial localization of *id2b* was further confirmed by RNAscope analysis (*Figure 2E*). In adult hearts, *id2b:eGFP* fluorescence was enriched in the chamber endocardium and the endothelium lining AVC, OFT, and bulbus arteriosus (*Figure 2F*). Interestingly, there was an absence of *id2b:eGFP* signal in *kdrl:mCherry⁺* endothelial cells in trunk blood vessel and brain vasculature (*Figure 2G*). Collectively, these results indicate that *id2b* is expressed in endocardial cells across different developmental stages.

## BMP signaling and cardiac contraction regulate *id2b* expression

Taking advantage of live imaging on developing embryos, we explored the in vivo dynamics of *id2b* in response to biomechanical force at single-cell resolution. When embryos were treated with tricaine or blebbistatin, the intensity of *id2b:eGFP* in atrial and ventricular endocardium was significantly reduced (*Figure 3A and B*). Similarly, injection of *tnnt2a* morpholino also markedly suppressed *id2b:eGFP* signal (*Figure 3A and B*), in agreement with the results obtained from in situ hybridization. Strikingly, the reduction in fluorescence intensity was particularly pronounced in AVC endothelial cells (*Figure 3A and B*, arrowheads).

We then explored how cardiac contraction modulated *id2b* expression. Given that endocardial cells can sense blood flow through primary cilia (*Li et al., 2020*; *Nauli et al., 2008*), we used a characterized morpholino (*Li et al., 2020*) to knock down *ift88*, an intraflagellar transporter essential for primary cilia formation. Previously, work demonstrated a complete loss of primary cilia in endocardial cells upon *ift88* knockdown (*Li et al., 2020*). As expected, a significant decrease in *id2b:eGFP* intensity was observed in the chamber and AVC endocardium of *ift88* morphants compared to control (*Figure 3C and D*), suggesting that biomechanical forces promote the expression of *id2b* via primary cilia. In the developing heart, a central hub for mediating biomechanical cues is the Klf2 gene, which includes the *klf2a* and *klf2b* paralogues in zebrafish (*Vermot et al., 2009*; *Heckel et al., 2015*; *Gálvez-Santisteban et al., 2019*; *Li et al., 2020*; *Rasouli et al., 2018*). Previous studies in mammals and zebrafish have highlighted the essential role of Klf2 TF activity in cardiac valve and myocardial wall formation (*Vermot et al., 2009*; *Rasouli et al., 2018*). As a flow-responsive gene, *klf2a* expression has been observed throughout the entire endocardium, evidenced by mRNA expression and transgenic studies (*Vermot et al., 2009*; *Heckel et al., 2015*). Interestingly, in situ hybridization on 48 and 72 hpf *klf2a⁻ᐟ⁻* and *klf2b⁻ᐟ⁻* embryos unveiled a drastic decrease in *id2b* expression compared with wild-type zebrafish (*Figure 3E*), supporting the notion that *klf2*-mediated biomechanical signaling is essential for activating *id2b* expression.

Given that *id2b* has been reported as a target gene of BMP signaling, we explored whether BMP also played a role in regulating *id2b* expression. To this end, we knocked down *bmp2b*, *bmp4*, and *bmp7a* in one-cell stage embryos. Live imaging at 24 hpf revealed a significant reduction in *id2b:eGFP* fluorescence signal in morpholino-injected hearts compared to controls (*Figure 3—figure supplement 1A and B*), suggesting that *id2b* is a target gene of BMP signaling during early embryonic development. Similarly, treatment with the BMP inhibitor Dorsomorphin from 10 to 24 hpf resulted in a marked decrease in *id2b:eGFP* signal (*Figure 3—figure supplement 1C and D*). Considering that heartbeats in zebrafish commence at approximately 22 hpf, we treated embryos with Dorsomorphin from 24 to 48 hpf or from 36 to 60 hpf. While the number of endocardial cells was slightly reduced upon Dorsomorphin exposure as previously reported (*Dietrich et al., 2014*), surprisingly, quantification of the average *id2b:eGFP* fluorescence intensity in individual endocardial cells revealed no significant differences between Dorsomorphin and DMSO-treated controls (*Figure 3—figure supplement 1C and D*).

We further visualized BMP activity using the *BRE:d2GFP* reporter line. Confocal images revealed strong fluorescence in the myocardium at 72 hpf, with minimal signal present in the endocardium except for the AVC endothelium (*Figure 3—figure supplement 1E*). Moreover, after tricaine treatment, endocardial *BRE:d2GFP* slightly increased (*Figure 3—figure supplement 1E*), as opposed to the reduced *id2b:eGFP* signal (*Figure 3A and B*). Likewise, endocardial *BRE:d2GFP* intensity was barely affected after completely blocking contraction with *tnnt2a* MO injection (*Figure 3—figure supplement 1E*). These observations align with previous work using pSmad-1/5/8 as a readout of BMP activity, indicating that endocardial BMP signaling is independent of blood flow (*Dietrich et al.,*

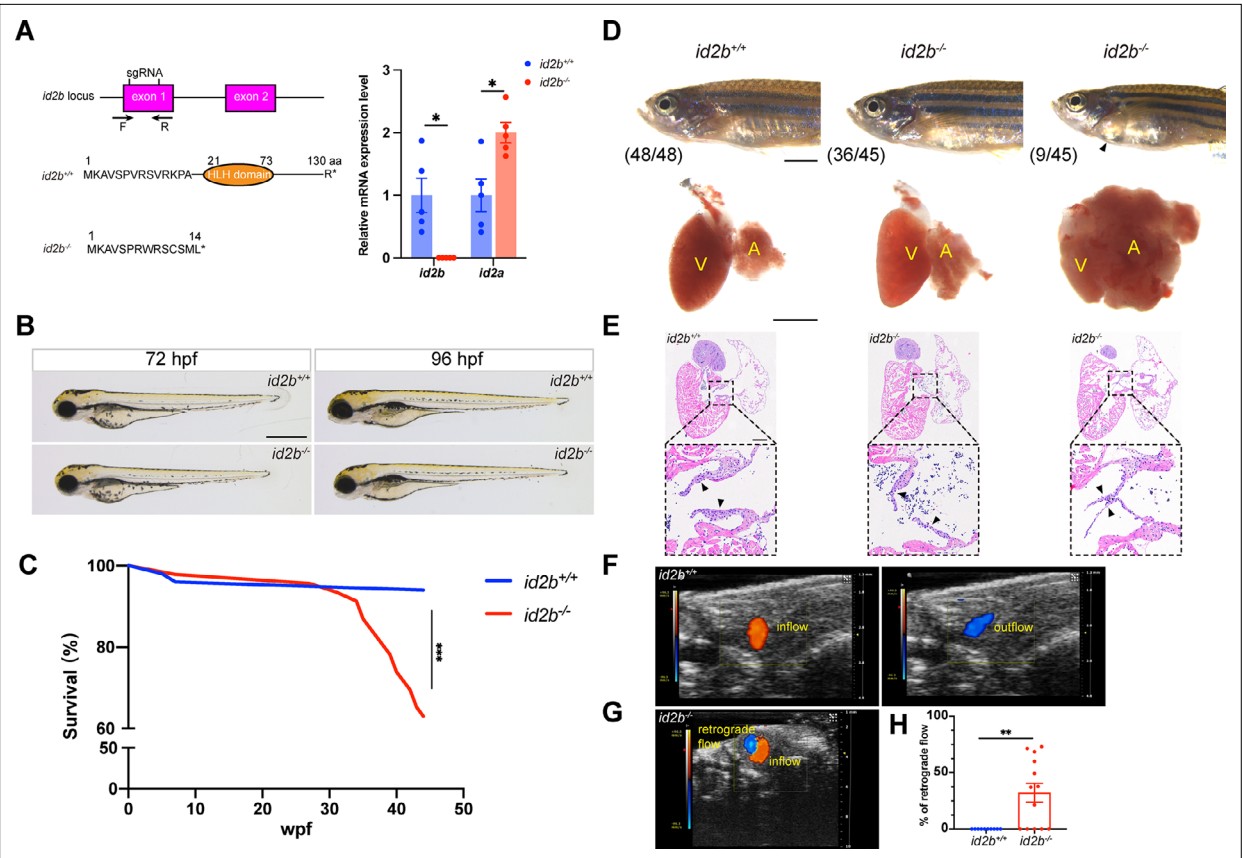

**Figure 4.** *id2b⁻/⁻* adults exhibit thinner atrioventricular valve leaflets and prominent retrograde blood flow. (**A**) Two sgRNAs, represented by short vertical lines, were designed to create *id2b⁻/⁻* mutants. Co-injection of the two sgRNAs with zCas9 protein induces a 157 bp truncation in the exon 1 of *id2b*, which can be detected by genotyping primers marked with arrows. This genetic modification leads to the formation of a premature stop codon and the subsequent loss of the helix-loop-helix (HLH) domain. Right, quantitative real-time PCR (qRT-PCR) analysis of *id2b* and *id2a* mRNA levels in *id2b⁺/⁺* and *id2b⁻/⁻* adult hearts. (**B**) No discernible morphological differences were observed between *id2b⁺/⁺* and *id2b⁻/⁻* larvae at both 72 hr post-fertilization (hpf) and 96 hpf. (**C**) Kaplan-Meier survival curve analysis and log-rank test of *id2b⁺/⁺* (n=50) and *id2b⁻/⁻* (n=46). Wpf, weeks post-fertilization. (**D**) Pericardial edema and an enlarged atrium are evident in a subset of *id2b⁻/⁻* adults. V, ventricle; A, atrium. (**E**) *id2b⁻/⁻* adults developed thinner atrioventricular valve leaflets (denoted by arrowheads) compared to *id2b⁺/⁺*. Enlarged views of boxed areas are shown in the bottom panels. (**F, G**) Echocardiograms of adult *id2b⁺/⁺* (**F**) and *id2b⁻/⁻* (**G**) hearts. Unidirectional blood flow was observed in the *id2b⁺/⁺* heart, while retrograde blood flow was evident in the *id2b⁻/⁻* heart. (**H**) Ratio of retrograde flow area over inflow area shows a significant increase in retrograde flow in *id2b⁻/⁻* (n=13) compared to *id2b⁺/⁺* (n=10). Data are presented as mean ± s.e.m. Unpaired two-tailed Student's t-tests were used to determine statistical significance. *p<0.05, **p<0.01, ***p<0.001. Scale bars, 500 μm (**B** and **D**, bottom), 2 mm (**D**, top), 200 μm (**E**).

The online version of this article includes the following figure supplement(s) for figure 4:

**Figure supplement 1.** id2b⁻/⁻ larvae exhibit a decreased number of valve endocardial cells while maintaining normal atrioventricular valve function.

*2014*). Collectively, these results suggest that *id2b* expression is regulated by both BMP and biomechanical signaling, with the relative contribution of each pathway varying across developmental stages.

## Compromised AV valve formation in *id2b* mutants

To investigate the role of the contractility-*id2b* axis in zebrafish heart development, we generated a loss-of-function mutant line using CRISPR/Cas9. A pair of sgRNAs designed to target exon 1 was injected with zCas9 protein into one-cell stage embryos. Consequently, we identified a mutant allele with a 157 bp truncation, leading to the generation of a premature stop codon (*Figure 4A*, left). In *id2b* mutants (*id2b⁻/⁻*), the expression levels of *id2b* were significantly decreased, while *id2a* expression levels were increased compared to *id2b⁺/⁺* siblings (*Figure 4A*, right). The overall morphology of *id2b⁻/⁻* remained unaltered at 72 and 96 hpf (*Figure 4B*). However, *id2b⁻/⁻* zebrafish experienced early lethality starting around 31 weeks post-fertilization (*Figure 4C*). Strikingly, pericardial edema was observed in 20% (9/45) of adult *id2b⁻/⁻* zebrafish (*Figure 4D*, top). Upon dissecting hearts from these

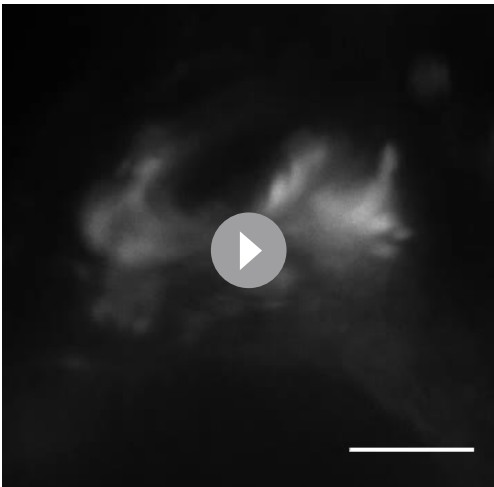

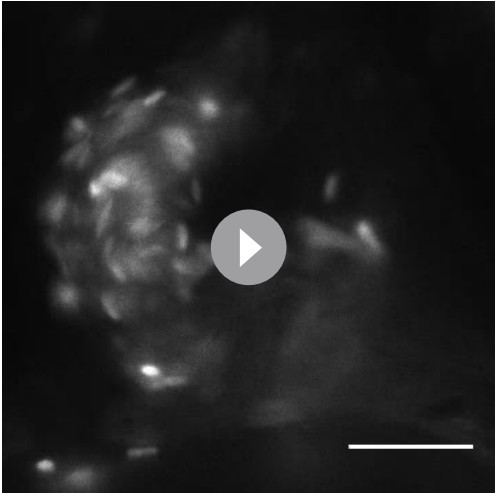

**Video 1.** 96 hr post-fertilization (hpf) *id2b⁺/⁺* larvae displayed unidirectional blood flow in the atrioventricular (AV) canal. Scale bar, 50 µm.
https://elifesciences.org/articles/101151/figures#video1

**Video 2.** 96 hr post-fertilization (hpf) *id2b⁻/⁻* larvae displayed unidirectional blood flow in the atrioventricular (AV) canal. Scale bar, 50 µm.
https://elifesciences.org/articles/101151/figures#video2

*id2b*⁻/⁻ zebrafish, a prominent enlargement in the atrium with a smaller ventricle was detected (*Figure 4D*, bottom), which has been characterized as cardiomyopathy in zebrafish (*Weeks et al., 2024*; *Kamel et al., 2021a*). Histological analysis further revealed malformation in the AV valves of these *id2b*⁻/⁻ mutants compared to controls (*Figure 4E*, right). Specifically, we noted that the superior and inferior leaflets were significantly thinner, comprising only one to two layers of cells in *id2b*⁻/⁻ zebrafish with an enlarged atrium. This was in sharp contrast to *id2b*⁺/⁺ zebrafish, which exhibited multilayers of cells (*Figure 4E*, left). Subsequent examination of the remaining 80% of *id2b*⁻/⁻ zebrafish (36/45) that did not display prominent pericardial edema also revealed AV valve malformation, albeit to a lesser extent (*Figure 4E*, middle).

To further interrogate the effect of *id2b* inactivation on AV valve formation and function, we analyzed the number of AVC endothelial cells using *kdrl:nucGFP*. At 96 hpf, a reduced number of *kdrl:nucGFP*⁺ cells were detected in the AVC region of *id2b*⁻/⁻ embryos compared with *id2b*⁺/⁺ (*Figure 4—figure supplement 1A and B*). In contrast, the number of atrial and ventricular endocardial cells did not differ between *id2b*⁻/⁻ and *id2b*⁺/⁺ (*Figure 4—figure supplement 1C and D*). Subsequently, we assessed hemodynamic flow by conducting time-lapse imaging of red blood cells labeled by *gata1:dsred*. Surprisingly, the pattern of hemodynamics was largely preserved in *id2b*⁻/⁻ embryos compared to *id2b*⁺/⁺ siblings at 96 hpf (*Figure 4—figure supplement 1E*, *Videos 1 and 2*), suggesting that the reduced number of endocardial cells in the AVC region was not sufficient to induce functional defects. Additionally, we performed echocardiography to analyze blood flow in adult zebrafish as previously described (*Gunawan et al., 2020*). In *id2b*⁻/⁻ hearts, prominent retrograde blood flow was detected in the AVC region (8/13) (*Figure 4G*), while unidirectional blood flow was observed in *id2b*⁺/⁺ (10/10) (*Figure 4F*). Quantification analysis showed ~32% retrograde blood flow in *id2b*⁻/⁻, compared to 0% in *id2b*⁺/⁺ zebrafish (*Figure 4H*). Consistently, the superior and inferior leaflets were much thinner in *id2b*⁻/⁻ exhibiting retrograde flow compared with control fish (*Figure 4—figure supplement 1F*). Overall, these histological and functional analyses indicate that *id2b* deletion leads to progressive defects in AV valve morphology and hemodynamic flow.

## *id2b* deletion perturbs calcium signaling and contractile function in the myocardium

Although similar defects in AV valve formation have been reported in both *klf2a* and *nfatc1* mutants, they do not display noticeable pericardial edema at the adult stage, nor do they experience early lethality (*Vermot et al., 2009*; *Li et al., 2020*; *Rasouli et al., 2018*; *Gunawan et al., 2020*; *Novodvorsky et al., 2015*). Therefore, we sought to investigate whether other cardiac properties have also been affected by *id2b* loss-of-function. To this end, we employed RNA-seq analysis on purified

embryonic *id2b*-/- and *id2b*+/+ hearts (*Figure 5—figure supplement 1*). As expected, enrichment analysis of DEGs demonstrated that the top-ranked anatomical structures affected by *id2b* deletion included the heart valve, the compact layer of ventricle, and the AVC (*Figure 5—figure supplement 1A*). Interestingly, *id2b* inactivation also impacted phenotypes such as cardiac muscle contraction and heart contraction (*Figure 5—figure supplement 1B*). Therefore, we investigated cardiac contractile

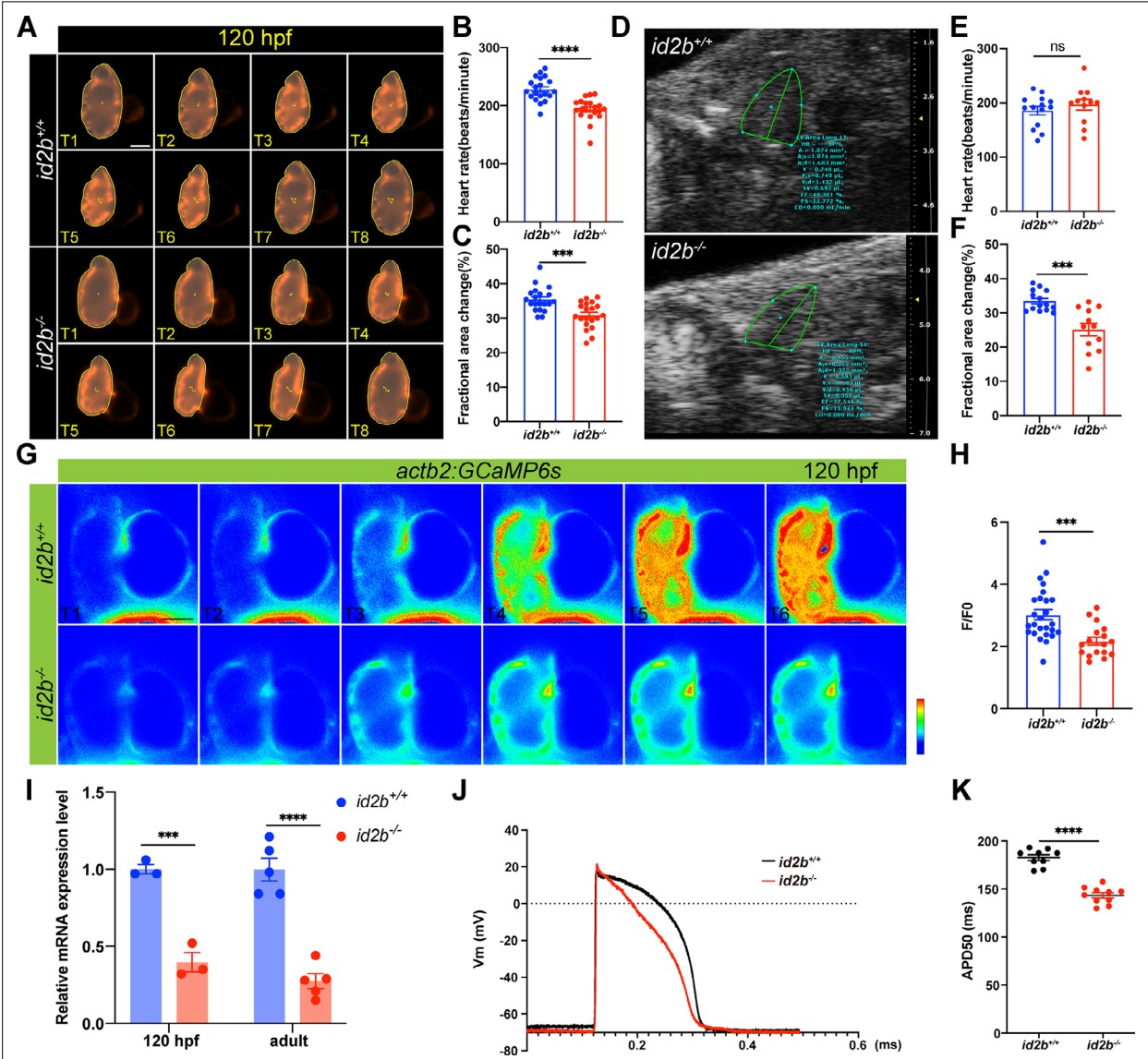

**Figure 5.** Reduced cardiac contractile function and compromised calcium handling in *id2b*-/- mutants. (**A**) Time-lapse imaging (from T1 to T8) illustrates the cardiac contraction-relaxation cycle of 120 hr post-fertilization (hpf) *id2b*+/+ and *id2b*-/- hearts carrying *myl7:mCherry*. (**B and C**) *id2b*-/- larvae (n=20) display a significant decrease in heart rate and fractional area change compared to *id2b*+/+ (n=20). (**D**) Echocardiograms of adult *id2b*+/+ and *id2b*-/- hearts. (**E and F**) *id2b*-/- fish (n=12) exhibit reduced cardiac contractile function with preserved heart rate compared to *id2b*+/+ (n=14). (**G**) Time-lapse imaging illustrates the calcium dynamics of 120 hpf *id2b*+/+ and *id2b*-/- hearts carrying *actb2:GCaMP6s*. (**H**) Ratio of maximal fluorescence intensity (**F**) over basal fluorescence intensity (**F0**) of GCaMP6s signal. n=(26, 17). (**I**) Quantitative real-time PCR (qRT-PCR) analysis of *cacnα1c* mRNA in *id2b*+/+ and *id2b*-/- hearts at 120 hpf (N=3 biological replicates, with each sample containing 500–1000 embryonic hearts) and adult stage (N=5 biological replicates). Data were normalized to the expression of *actb1*. (**J**) The action potential of ventricular cardiomyocytes in adult *id2b*+/+ (n=9) and *id2b*-/- (n=9) hearts. (**K**) Statistical data showed a notable difference between *id2b*+/+ and *id2b*-/- accordingly. Data are presented as mean ± s.e.m. p-values were calculated by unpaired two-tailed Student's t-tests. ***p<0.001, ****p<0.0001, ns, not significant. Scale bars, 50 μm.

The online version of this article includes the following figure supplement(s) for figure 5:

**Figure supplement 1.** id2b loss-of-function impacts both valve formation and cardiac contraction.

**Figure supplement 2.** id2b-/- hearts develop normal trabeculae and sarcomeres.

function through time-lapse imaging on the *myl7:mCherry* background. At 72 and 120 hpf, a significant decrease in cardiac function was observed in *id2b*⁻/⁻ compared with *id2b*⁺/⁺ (*Figure 5A–C*, *Figure 5— figure supplement 2A–C*). Similarly, echocardiography analysis showed that the contractile function in adult *id2b*⁻/⁻ heart was dramatically reduced compared with age-matched *id2b*⁺/⁺ (*Figure 5D and F*). These functional defects in *id2b*-deleted hearts could not be attributed to differences in cardiomyocyte number, as we counted cardiomyocytes using the *myl7:H2A-mCherry* line and found no apparent changes between *id2b*⁻/⁻ and *id2b*⁺/⁺ embryos at 72 and 120 hpf (*Figure 5—figure supplement 2D and E*). Similarly, *id2b*⁻/⁻ also developed regular trabecular structures (*Figure 5—figure supplement 2F*). Through α-actinin immunostaining, we observed similar sarcomeric structures in *id2b*⁻/⁻ and control cardiomyocytes at 72 hpf and adult stages (115 dpf) (*Figure 5—figure supplement 2G*), corroborating that the reduced contractility in *id2b*-depleted heart was independent of structural defects.

The key functional unit that transmits electrical activity to contractile function is E-C coupling. Because *id2b*⁻/⁻ displayed reduced cardiac function, we visualized calcium signaling in the developing heart using *actb2:GCaMP6s* zebrafish (*Figure 5G*). Compared to *id2b*⁺/⁺ controls, *id2b*⁻/⁻ embryos exhibited markedly decreased calcium transient amplitude (*Figure 5H*), consistent with compromised calcium handling observed in other zebrafish cardiomyopathy models (*Kamel et al., 2021a*; *Kamel et al., 2021b*). In cardiomyocyte, the entry of extracellular calcium is mainly mediated through the LTCC. As previously reported, a defect in zebrafish LTCC pore-forming α1 subunit *cacna1c* leads to compromised cardiac function (*Rottbauer et al., 2001*). We collected hearts from 72 hpf and 5 months post-fertilization zebrafish and detected downregulated *cacna1c* in *id2b*⁻/⁻ compared to *id2b*⁺/⁺ (*Figure 5I*). In addition, we measured cardiac action potential using intracellular recording (*Zhang et al., 2013*). Compared to *id2b*⁺/⁺ zebrafish, the duration of the action potential in *id2b*⁻/⁻ was significantly shorter (*Figure 5J and K*), consistent with the decreased expression level of *cacna1c*. Together, these data indicate that *id2b* loss-of-function leads to compromised calcium signaling and cardiac contractile function.

## Reduced expression of *nrg1* mediates the compromised contractility in *id2b*⁻/⁻

Because the deficiency of *id2b* in the endocardium disrupted the function of myocardium, we speculated that the cross talk between these two types of cells was affected in *id2b*⁻/⁻. Interestingly, comparing the DEGs in embryonic *id2b*⁻/⁻ and *id2b*⁺/⁺ hearts identified a significant reduction in the expression level of Nrg1, a key mitogen regulating the intra-organ communications between endocardial cells and cardiomyocytes (*Figure 6A*). Remarkably, analysis of a zebrafish single-cell database (*Jiang et al., 2021*) revealed enriched expression of *nrg1* in endocardial cells (*Figure 6—figure supplement 1*). However, attempts to detect *nrg1* expression through in situ hybridization were unsuccessful, likely due to its low abundance in the heart. Alternatively, qRT-PCR analysis of purified 120 hpf embryonic hearts validated decreased *nrg1* levels in *id2b*⁻/⁻ compared to control (*Figure 6B*). Previous studies have demonstrated that perturbations in Nrg-Erbb2 signaling, as seen in zebrafish *erbb2* mutants, result in dysfunctional cardiac contractility (*Liu et al., 2010*). Consistently, a decrease in heart rate was observed in embryos treated with the *erbb2* inhibitor AG1478 (*Figure 6C*).

Remarkably, injecting *nrg1* mRNA at the one-cell stage not only rescued the reduced expression of *cacna1c* in *id2b*⁻/⁻ hearts (*Figure 6D*) but also restored the diminished heart rate (*Figure 6E*). This is consistent with prior studies showing that Nrg1 administration can restore LTCC expression and calcium current in failing mammalian cardiomyocytes (*Wang et al., 2019*). Overall, our data suggest that endocardial *id2b* promotes Nrg1 synthesis, thereby enhancing cardiomyocyte contractile function.

## Id2b interacts with Tcf3b to limit its repressor activity on *nrg1* expression

We further interrogated how *id2b* promotes the expression of *nrg1*. As an HLH factor lacking a DNA-binding motif, Id2b has been reported to form a heterodimer with Tcf3b to limit its function as a potent transcriptional repressor (*Slattery et al., 2008*). Notably, we detected expression of *tcf3b* in endocardial cells by analyzing a zebrafish single-cell database (*Jiang et al., 2021*; *Figure 7—figure supplement 1*). To determine if zebrafish Id2b and Tcf3b interact in vitro, Flag-*id2b* and HA-*tcf3b* were co-expressed in HEK293 cells. Co-immunoprecipitation analysis confirmed their interaction (*Figure 7A*), although whether they interact in vivo remains to be further investigated. Subsequently,

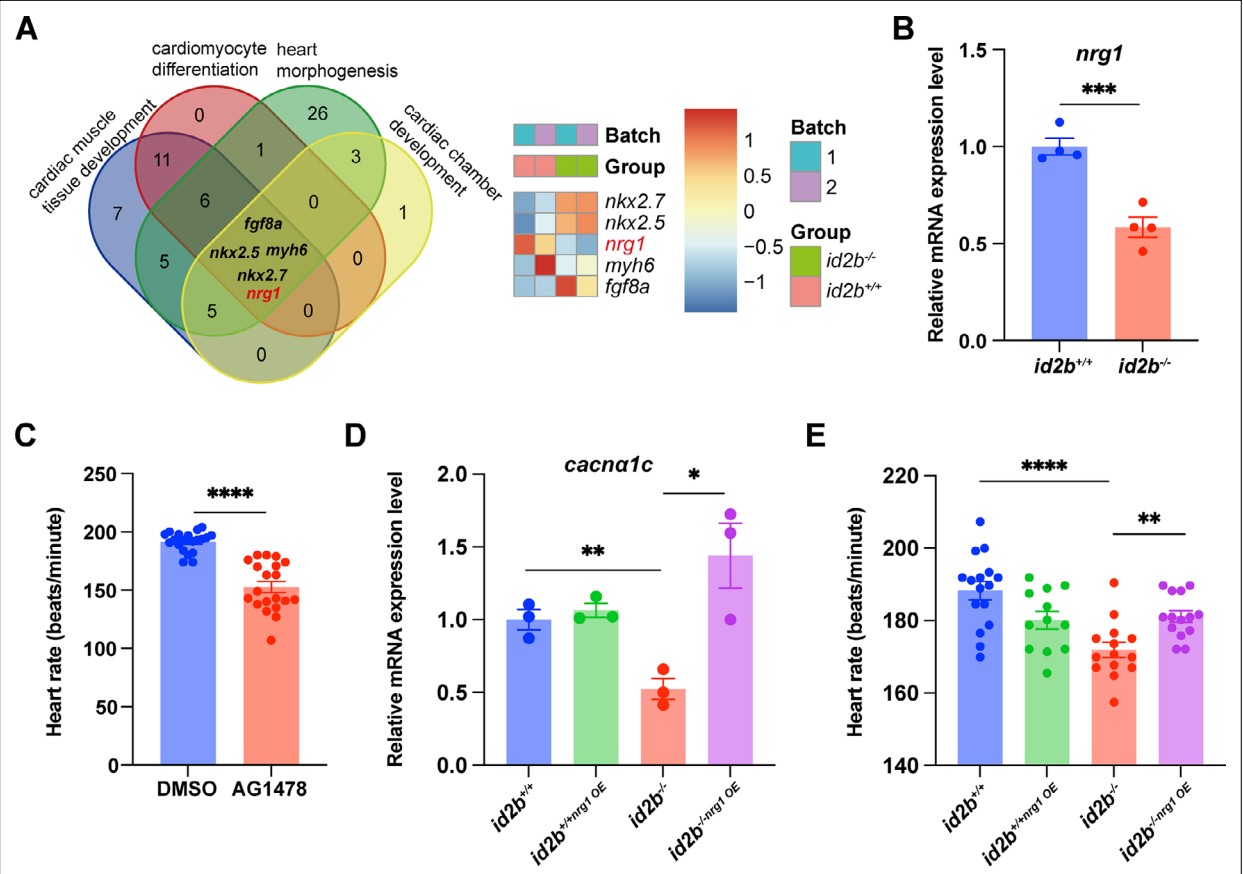

**Figure 6.** Nrg1 serves as a pivotal mitogen mediating the function of *id2b*. (**A**) Identification of genes (*fgf8a*, *nkx2.5*, *myh6*, *nrg1*, and *nkx2.7*) associated with four distinct heart development processes: cardiac muscle tissue development, cardiomyocyte differentiation, heart morphogenesis, and cardiac chamber development. The heatmap illustrates scaled-normalized expression values for the mentioned genes. (**B**) Quantitative real-time PCR (qRT-PCR) analysis of *nrg1* mRNA in 120 hr post-fertilization (hpf) *id2b⁺/⁺* and *id2b⁻/⁻* embryonic hearts. Data were normalized to the expression of *actb1*. N=4 biological replicates, with each sample containing 500–1000 embryonic hearts. (**C**) Heart rate in 120 hpf larvae treated with AG1478 (n=20) and DMSO (n=20). (**D**) *id2b⁺/⁺* and *id2b⁻/⁻* larvae were injected with *nrg1* mRNA at the one-cell stage, followed by qRT-PCR analysis of *cacnα1c* mRNA at 72 hpf. Data were normalized to the expression of *actb1*. N=3 biological replicates, with each sample containing 100–200 embryonic hearts. (**E**) The heart rate of 72 hpf *id2b⁺/⁺* and *id2b⁻/⁻* larvae injected with *nrg1* mRNA at one-cell stage. n=(16, 12, 14, 14). Data are presented as mean ± s.e.m. p-values were calculated by unpaired two-tailed Student's t-tests. *p<0.05, **p<0.01, ***p<0.001, ****p<0.0001.

The online version of this article includes the following figure supplement(s) for figure 6:

**Figure supplement 1.** *nrg1* is expressed in the endocardial cells.

qRT-PCR analysis on purified 120 hpf embryonic hearts revealed a significant increase in the expression of *socs3b* and *socs1a*, target genes of *tcf3b*, in *id2b⁻/⁻* compared to *id2b⁺/⁺* (*Figure 7B*). This suggests an elevation in *tcf3b* activity associated with the loss of *id2b* function. Notably, the expression levels of *tcf3a* and *tcf3b* remained consistent between *id2b⁻/⁻* and *id2b⁺/⁺* hearts (*Figure 7B*).

To understand how the altered interaction between *id2b* and *tcf3b* influences *nrg1* expression, we analyzed the promoter region of zebrafish *nrg1* using JASPAR and identified two potential *tcf3b* binding sites (*Figure 7C*). Subsequently, a DNA fragment containing the zebrafish *nrg1* promoter region was subcloned into a vector carrying the luciferase reporter gene. Co-injection of this construct with *tcf3b* mRNA into one-cell stage embryos resulted in a significant decrease in luciferase signal. Conversely, co-injection with a previously characterized *tcf3b* morpholino led to enhanced luciferase intensity (*Figure 7D*). These results suggest a possible mechanism by which Tcf3b represses *nrg1* expression in zebrafish.

Lastly, injecting *tcf3b* morpholino into *id2b⁻/⁻* embryos was performed to assess whether attenuating the overactive *tcf3b* in *id2b⁻/⁻* could restore the expression level of *nrg1*. qRT-PCR analysis of purified 72 hpf hearts revealed a partial restoration of the diminished *nrg1* expression in *id2b⁻/⁻* upon *tcf3b*

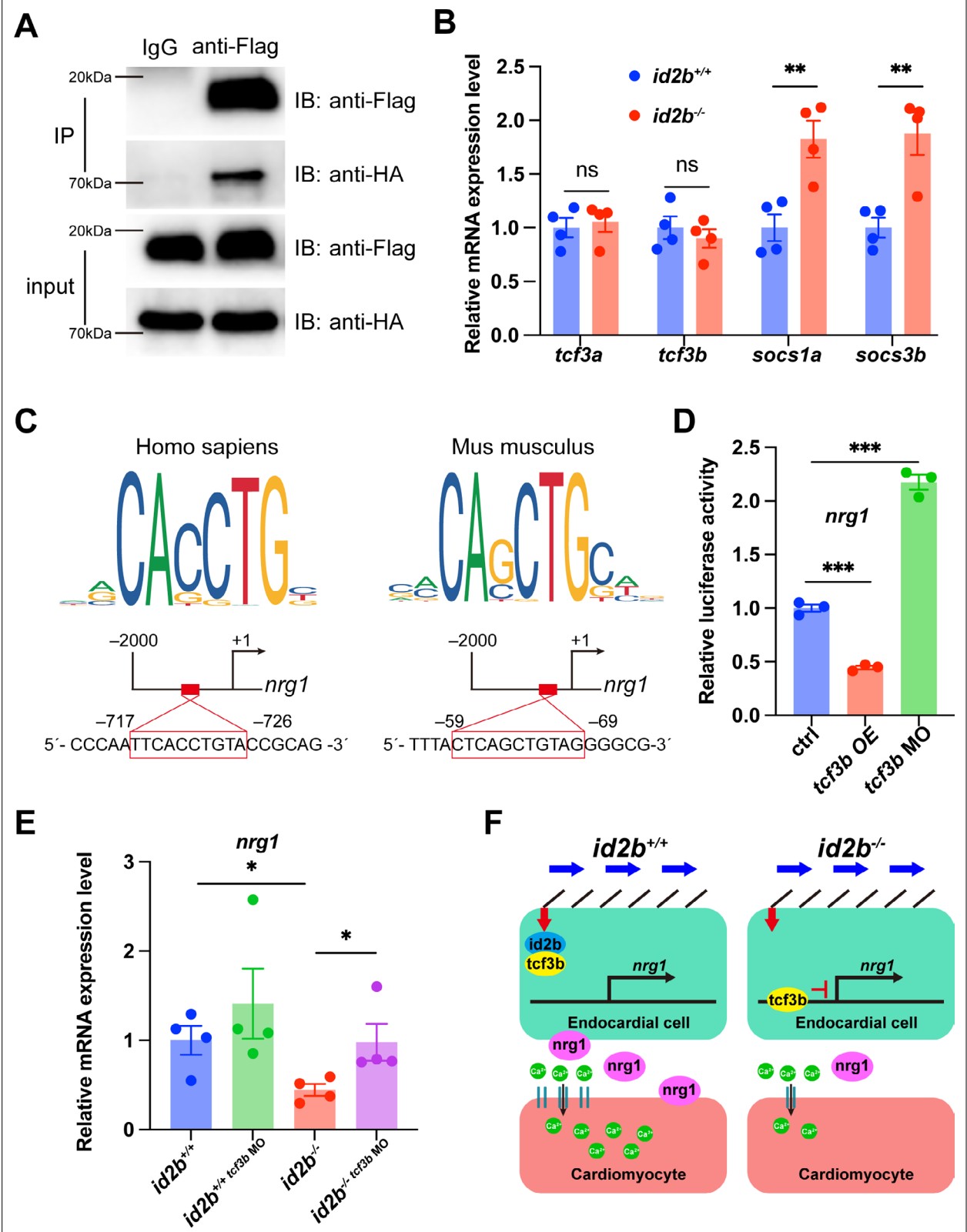

**Figure 7.** Id2b interacts with Tcf3b to restrict its inhibition on *nrg1* expression. (**A**) Immunoprecipitation (IP) assays of Flag-*id2b* and HA-*tcf3b* co-transfected 293T cells. (**B**) Quantitative real-time PCR (qRT-PCR) analysis of *tcf3a*, *tcf3b*, *socs1a*, and *socs3b* mRNA in 120 hr post-fertilization (hpf) *id2b*^+/+^ and *id2b*^-/-^ embryonic hearts. Data were normalized to the expression of *actb1*. N=4 biological replicates, with each sample containing 500–1000 embryonic hearts. (**C**) Two potential Tcf3b-binding sites, with sequences corresponding to the human TCF3 (left) and mouse Tcf3 (right) binding motifs,

*Figure 7 continued on next page*

*Figure 7 continued*

were predicted in the 2000 bp DNA sequence upstream of the zebrafish *nrg1* transcription start site using JASPAR. (**D**) Luciferase assay showing the expression of *nrg1* in embryos with *tcf3b* overexpression (*tcf3b* OE) and morpholino-mediated *tcf3b* knockdown (*tcf3b* MO). N=3 biological replicates. (**E**) qRT-PCR analysis of *nrg1* mRNA in 72 hpf *id2b*^+/+^ and *id2b*^-/-^ embryonic hearts injected with control and *tcf3b* morpholino. Data were normalized to the expression of *actb1*. N=4 biological replicates, with each sample containing 100–200 embryonic hearts. (**F**) Schematic model for *id2b*-mediated regulation of myocardium function. During heart development, blood flow operates through primary cilia, initiating endocardial *id2b* expression. Subsequently, the interaction between Id2b and Tcf3b restricts the activity of Tcf3b, ensuring proper *nrg1* expression, which in turn promotes L-type calcium channel (LTCC) expression (left). However, in the absence of Id2b, Tcf3b inhibits *nrg1* expression. The reduced Nrg1 hinders LTCC expression in cardiomyocytes, resulting in decreased extracellular calcium entry and disruption of myocardial function. Data are presented as mean ± s.e.m. p-values were calculated by unpaired two-tailed Student's t-tests. *p<0.05, **p<0.01, ***p<0.001. ns, not significant.

The online version of this article includes the following source data and figure supplement(s) for figure 7:

**Source data 1.** Source data for western blot shown in *Figure 7A*.

**Source data 2.** Original files for western blot shown in *Figure 7A*.

**Figure supplement 1.** Expression landscape of zebrafish *tcf3b*.

inhibition (*Figure 7E*). Taken together, our results indicate that biomechanical cues activate endocardial Id2b expression, leading to its interaction with Tcf3b to alleviate repression on the *nrg1* promoter. Consequently, the depletion of *id2b* unleashes Tcf3b's repressor activity, leading to a reduction in *nrg1* expression, which further acts through *erbb2* to regulate cardiomyocyte function (*Figure 7F*).

## Discussion

Biomechanical forces play an essential role in regulating the patterning and function of the heart. At AVC, oscillatory flow promotes the expression of *klf2a* and *nfatc1* to modulate valve morphogenesis. In chamber endocardium, blood flow induces endocardial cells to acquire distinct cell morphology. However, it still lacks a systematic analysis of the transcriptome underlying compromised heartbeats. In the present study, we analyzed embryonic zebrafish hearts without contractility and identified genes that are regulated by biomechanical forces. Specifically, our results unveiled the endocardial-specific expression of *id2b*, which was tightly regulated by flow-sensitive primary cilia-*klf2* axis. Genetic deletion of *id2b* resulted in compromised valve formation and progressive atrium enlargement. In addition, a reduction in heart rate and contractile force was observed in *id2b*^-/-^, owing to decreased expression of LTCC α1 subunit *cacna1c*. Mechanistically, *id2b* interacts with bHLH TF *tcf3b* to limit its repressor activity. Hence, genetic deletion of *id2b* unleashes *tcf3b* activity, which further represses endocardial *nrg1* expression. As a result, injection of *nrg1* mRNA partially rescues the phenotype of *id2b* deletion. Overall, our findings identify *id2b* as a novel mediator that regulates the interplay between endocardium and myocardium during heart development.

In mammals, the deletion of *Id2* leads to malformations in the arterial and venous poles of the heart, as well as affects AV valve morphogenesis (*Jongbloed et al., 2011*; *Moskowitz et al., 2011*). Interestingly, approximately 20% of perinatal lethality is reported in *Id2* knockout mice, exhibiting AV septal defects and membranous ventricular septal defects (*Moskowitz et al., 2011*). Remarkably, pericardial edema is evident in 20% of adult *id2b*^-/-^ zebrafish, with a prominent enlargement of the atrium. The superior and inferior leaflets of AV valves in *id2b*^-/-^ mutants are significantly thinner compared to the control. Therefore, our results suggest that *id2b* may play a similar role in regulating AV valve formation in zebrafish as its mammalian orthologue *Id2*. It is proposed that the loss of *Id2* in mice results in compromised endocardial proliferation and aberrant endothelial-to-mesenchymal transformation, collectively leading to defective valve morphogenesis (*Moskowitz et al., 2011*). Nevertheless, the mechanism by which *id2b* loss-of-function causes a reduction in leaflet thickness in zebrafish remains to be determined in future studies.

*id2b* has been recognized as a target gene of the BMP signaling pathway. As expected, knockdown of *bmp2b*, *bmp4*, and *bmp7a* at one-cell stage confirms that endocardial *id2b* expression is controlled by BMP activity during early embryonic development. Surprisingly, treatment with the BMP inhibitor Dorsomorphin at 24 and 36 hpf, when cardiac contractions have already initiated, fails to alter *id2b* expression in the endocardium, suggesting that BMP is dispensable for *id2b* activation at these stages. Instead, endocardial *id2b* expression is reduced upon loss-of-function of *klf2a*, *klf2b*, and *ift88*, suggesting an essential role of the primary cilia-*klf2* axis in mediating *id2b* activation. In

endocardial cells, Trp, Piezo, and ATP-dependent P2X/P2Y channels (*Heckel et al., 2015*; *Fukui et al., 2021*; *Li et al., 2014*; *Nonomura et al., 2018*) are well-established sensors for biomechanical stimulation. The activation of these channels further promotes the activities of Klf2 and Nfatc1 to drive heart development and valvulogenesis. However, whether these channels are also required for the activation of *id2b* warrants further investigation.

The Nrg-Erbb signaling plays an essential role in regulating heart morphogenesis and function. In the mammalian heart, the genetic deletion of Nrg1 or Erbb2 results in severely perturbed cardiac trabeculae formation (*Gassmann et al., 1995*; *Lee et al., 1995*; *Meyer and Birchmeier, 1995*). Zebrafish *erbb2* mutants exhibit a similar defect in cardiomyocyte proliferation and trabeculation (*Liu et al., 2010*). Interestingly, *nrg1* mutant zebrafish display grossly normal cardiac structure during early embryonic development (*Rasouli and Stainier, 2017*; *Brown et al., 2018*). Nevertheless, zebrafish *nrg2a* loss-of-function leads to defective trabeculae formation, suggesting that *nrg2a* is the predominant ligand secreted from endocardium, promoting ventricular morphogenesis (*Rasouli and Stainier, 2017*). In the adult stage, perivascular cells (*Gemberling et al., 2015*) or regulatory T cell-derived (*Hui et al., 2017*) *nrg1* promotes cardiomyocyte proliferation during heart regeneration. Hence, the specific ligand/receptor and the spatiotemporal regulation of the Nrg-Erbb axis appear to be more complicated in both embryonic and adult zebrafish. Interestingly, the *nrg1* mutant heart exhibits a defect in cardiac nerve expansion and heart maturation at the juvenile stage despite normal cardiac structure formation (*Brown et al., 2018*), suggesting its potential role in regulating cardiac function. Our findings demonstrate that the expression of *nrg1* in embryonic endocardial cells is influenced by biomechanical cues and *id2b* activity. This signaling axis is essential for coordinating endocardium-myocardium interaction and establishing proper cardiac function.

## Materials and methods
### Zebrafish handling and lines

All animal procedures were approved by the Animal Care and Use Committee of the Zhejiang University School of Medicine (application no. 29296). Embryonic and adult fish were raised and maintained under standard conditions at 28°C on a 14/10 hr day/night cycle. The following zebrafish lines were used in this study: *Tg(myl7:mCherry)$^{sd7}$* (*Palencia-Desai et al., 2011*), *Tg(myl7:H2A-mCherry)$^{sd12}$* (*Schumacher et al., 2013*), *Tg(kdrl:mCherry)$^{S896}$* (*Chi et al., 2008*), *Tg(kdrl:nucGFP)$^{y7}$* (*Roman et al., 2002*), *Tg(BRE:d2GFP)$^{mw30}$* (*Collery and Link, 2011*), and *Tg(actb2:Gcamp6s)*. To generate the *id2b* mutant, two short guide RNAs (sgRNAs) targeting exon 1 were generated using the MAXIscript T7 transcription kit (ambion, AM1314). The sgRNAs were as follows: sgRNA1: 5' - GAAGGCAGTCAG TCCGGTG - 3'; sgRNA2: 5' - GAACCGGAGCGTGAGTAAGA - 3'. The two sgRNAs, along with zCas9 protein, were co-injected into one-cell stage embryos. Embryos were raised to adulthood and crossed to wild-type zebrafish to obtain $F_1$ progenies. Through PCR analysis, a mutant line with a 157 bp truncation was identified.

The knock-in *id2b:eGFP* line was generated using a previously reported method (*Li et al., 2015*). Briefly, three sgRNAs were designed to target the intron of *id2b* (sgRNA1: 5' - GAGACAAATATCTACT AGTG - 3'; sgRNA2: 5' - GTTGAACACATGACGATATT - 3'; sgRNA3: 5' - GCACAACTTAGATTTCAAGT - 3'). Co-injection of each individual sgRNA with zCas9 protein into one-cell stage zebrafish embryos yielded varying cleavage efficiency. Since sgRNA2 displayed the highest gene editing efficiency, it was selected for subsequent studies. Next, a donor plasmid containing the sgRNA targeting sequence of the intron, exon 2 of *id2b*, and P2A-eGFP was generated. Co-injection of sgRNA, donor plasmid, and zCas9 protein into one-cell stage embryos led to concurrent cleavage of the sgRNA targeting sites in both the zebrafish genome and the donor plasmid (*Figure 2A*). Accordingly, eGFP fluorescence was observed in injected 24 hpf zebrafish embryos, indicating the incorporation of the donor. The insertion of the *id2b*-p2A-eGFP donor into the genome was confirmed by PCR analysis with primers recognizing target site or donor sequences, respectively (*Figure 2A*). Embryos with mosaic eGFP expression were raised to adulthood and crossed with wild-type zebrafish to obtain $F_1$ progenies. Overall, two founders were identified. The junction region of the $F_1$ embryos was sequenced to determine the integration sites. Although the two founders had slightly different integration sites in the intron, the expression pattern and fluorescence intensity of eGFP were indistinguishable between the two lines.

## Morpholinos

All morpholinos (Gene Tools) used in this study have been previously characterized. *tnnt2a* MO (5' - CATGTTTGCTCTGATCTGACACGCA - 3') (*Sehnert et al., 2002*); *ift88* MO (5' - CTGGGACAAGAT GCACATTCTCCAT - 3') (*Li et al., 2020*); *bmp2b* MO (5' - ACCACGGCGACCATGATCAGTCAGT - 3') (*Lele et al., 2001*); *bmp4* MO (5' - AACAGTCCATGTTTGTCGAGAGGTG - 3') (*Weber et al., 2008*); *bmp7a* MO (5' - GCACTGGAAACATTTTTAGAGTCAT - 3') (*Lele et al., 2001*); *tcf3b* MO (5' - CGCC TCCGTTAAGCTGCGGCATGTT - 3') (*Dorsky et al., 2003*). For each morpholino, a 1 nL solution was injected into one-cell stage embryos at the specified concentrations: 0.5 µg/µL *tnnt2a* MO, 2 µg/µL *ift88* MO, 0.5 µg/µL *bmp2b* MO, 2 µg/µL *bmp4* MO, 4 µg/µL *bmp7a* MO, and 1 µg/µL *tcf3b* MO.

## Small molecules treatment

To inhibit cardiac contraction, embryos were incubated in 1 mg/mL tricaine (Sigma, A5040) or 10 µM blebbistatin (MedChemExpress, HY13441) PTU-added egg water for 12–24 hr. In order to inhibit *erbb2* signaling pathway, 10 µM AG1478 (Sigma, 658552) was used to treat 4 dpf larvae. To inhibit BMP signaling pathway, 10 µM Dorsomorphin (Sigma, P5499) was used to treat 10, 24, and 36 hpf embryos.

## In situ hybridization and RNAscope

Whole-mount in situ hybridization was performed as previously described (*Zhang et al., 2013*). The probes were synthesized using the DIG RNA labeling kit (Roche). The primers used for obtaining the *id2b* probe template were as follows: Forward 5' - ATGAAGGCAGTCAGTCCGGTGAGGT - 3'; Reverse 5' - TCAACGAGACAGGGCTATGAGGTCA - 3'. RNAscope analysis was performed using the probe Dr-*id2b* (Advanced Cell Diagnostics, 517541) and the Multiplex Fluorescent Detection Kit version 2 (Advanced Cell Diagnostics, 323100) as previously described (*Liang et al., 2025*).

## Embryonic heart isolation and RNA-seq analysis

Hearts were isolated from embryos carrying the *Tg*(*myl7:mCherry*) transgene following an established protocol (*Burns and MacRae, 2006*). A minimum of 1000 hearts for each experimental group was manually collected under a Leica M165FC fluorescence stereomicroscope and transferred into ice-cold PBS buffer. After centrifugation at 12,000×*g* for 2 min at 4°C, the supernatant was removed, and hearts were lysed in cold TRIzol buffer (Ambion, 15596). Total RNA was extracted for subsequent qRT-PCR or RNA-seq analysis.

Duplicate samples from control and Tricaine-treated embryonic hearts underwent RNA-seq. Raw sequencing reads were preprocessed to remove adapters and filter low-quality reads. Clean sequencing reads were then mapped to the zebrafish reference (*Weinberger et al., 2020*) using STAR with default parameters (*Dobin et al., 2013*). Subsequently, gene quantification was carried out with RSEM (*Li and Dewey, 2011*). The gene expected count was applied to identify DEGs, retaining only genes with counts per million of 10 in at least two samples. DESeq2 (*Love et al., 2014*) was employed for differential expression analysis, and p-values were adjusted using BH correction. DEGs were defined as those with |log$_2$fold change|≥0.585 and an adjusted p-value<0.1. The primary focus was on genes related to transcription regulation, and gene enrichment analysis was conducted using ClusterProfiler (*Wu et al., 2021*). To analyze DEGs in *id2b*[-/-] and control embryonic hearts, we performed enrichment analysis with the R package EnrichR, dissecting the potential anatomy expression pattern and underlying phenotypes. We mainly focused on genes with the heart-related phenotypes, including cardiac muscle tissue development, cardiomyocyte differentiation, heart morphogenesis, and cardiac chamber development. All the analysis on identifying DEGs was batch-corrected.

## qRT-PCR analysis

After extraction from isolated embryonic hearts, 50 ng to 1 µg of mRNA was reverse-transcribed to cDNA using the PrimeScript RT Master Mix kit (Takara, RR036A). Real-time PCR was performed using the TB Green Premix Ex Taq kit (Takara, RR420A) on the Roche LightCycler 480. Expression levels of the target genes were normalized to *actb1* as an internal control. All experiments were repeated three times. The following primer sets were used: *id2b* Forward 5' - ACCTTCAGATCGCACTGGAC - 3', Reverse 5' - CTCCACGACCGAAACACCATT - 3'; *nrg1* Forward 5' - CTGCATCATGGCTGAGGTGA - 3', Reverse 5' - TTAACTTCGGTTCCGCTTGC - 3'; *cacnα1c* Forward 5' - GCCCTTATTGTAGTGGGTAG

TG - 3', Reverse 5' - AGTGTTTTGGAGGCCCATTG - 3'; *tcf3a* Forward 5' - CCTCCGGTCATGAGCA ACTT - 3', Reverse 5' - TTTCCCATGATGCCTTCGCT - 3'; *tcf3b* Forward 5' - CCTTTAATGCGCCGTG CTTC - 3', Reverse 5' - GCGTTCTTCCATTCCTGTACCA - 3'; *socs1a* Forward 5' - TCAGCCTGACAG GAAGCAAG - 3', Reverse 5' - GTTGCACAGGGATGCAGTCG - 3'; *socs3b* Forward 5' - GGGACAGT GAGTTCCTCCAA - 3', Reverse 5' - ATGGGAGCATCGTACTCCTG - 3'; *actb1* Forward 5' - ACCA CGGCCGAAAGAGAAAT - 3', Reverse 5' - GCAAGATTCCATACCCAGGA - 3'.

## Co-IP and western blot

Zebrafish *tcf3b* and *id2b* were overexpressed in 293T cell for 48 hr. The transfected cells were then collected and lysed using IP lysis buffer (Sangon Biotech, C500035) containing protease and phosphatase inhibitors (Sangon Biotech, C510009, C500017). For the IP experiment, anti-Flag antibody (Cell Signaling Technology, 14793, 1:100) and IgG antibody (ABclonal, AC005, 1:100) were incubated with cell lysates overnight at 4°C. Pretreated magnetic beads were bound with the antigen-antibody complex for 4 hr at 4°C, followed by washing with IP lysis buffer three times. For western blot, samples were denatured at 95°C for 10 min, separated on a 5–12% gradient gel. Proteins were then transferred to a PVDF membrane (Sigma, ISEQ00010). The membrane was blocked for 1 hr with 5% nonfat milk or 5% BSA (Sangon Biotech) dissolved in TBST and then incubated with primary antibodies (anti-FLAG, Cell Signaling Technology, 14793, 1:1000; anti-HA, Sigma, H3663, 1:1000) overnight at 4°C, followed by three times 10 min TBST washes. HRP-conjugated secondary antibodies (Invitrogen, 31430, 31460) were incubated for 1 hr at room temperature, followed by three times 10 min TBST washes. The detection of immunoreactive bands was performed with a chemiluminescent substrate (Thermo Scientific, 34577) and imaged using the Azure Biosystems 400.

## Immunofluorescence

For immunofluorescence on adult zebrafish hearts, we fixed the hearts overnight in 4% paraformaldehyde at 4°C, followed by equilibration through 15% and 30% sucrose in PBS solution. The hearts were embedded and frozen in O.C.T. compound (Epredia, 6502), and 10-μm-thick cryosections were prepared using a CryoStar NX50 cryostat. Immunofluorescence experiments were performed as previously described (*Han et al., 2014*). For immunofluorescence on embryonic hearts, embryos were fixed overnight in 4% paraformaldehyde at 4°C, washed twice quickly in 100% methanol, and then dehydrated overnight at –20°C in 100% methanol. Subsequently, rehydration was performed through a methanol gradient (100%, 75%, 50%, 25%, 10 min each), followed by three times washes in PBST (1% PBS/0.1% Triton X-100, 10 min each). The embryos were treated with 10 μg/mL proteinase K diluted in PBST for 20 min at room temperature, refixed in 4% paraformaldehyde for 20 min, washed in PBST, and immersed in blocking solution (PBST/1% BSA/2% goat serum) for 1 hr at room temperature. Following this, the embryos were incubated in the primary antibody diluted in blocking solution overnight at 4°C. After washing in PBST, they were incubated in the secondary antibody (1:200) for 2 hr at room temperature. The primary antibody used was anti-GFP antibody (Santa Cruz Biotechnology, sc9996, 1:200) and anti-α-actinin antibody (Sigma, A7811, 1:200). The secondary antibody used was anti-mouse IgG-Alexa 488 (Invitrogen, A11011, 1:400). DAPI was used to stain cell nuclei.

## Cardiac function analysis

To assess cardiac function in embryonic hearts, embryos were incubated in 0.16 mg/mL tricaine (Sigma, A5040) and embedded in 1% low melting agarose. Heart contractions were recorded for 1 min using a Nikon Ti2 microscope at a rate of 25 frames per second. Fractional shortening and heart rate were measured as described previously (*Zhang et al., 2013*). For cardiac contractile functions in adulthood, zebrafish were fixed on a sponge soaked with system water with the belly facing up, and echocardiography was performed (*Wang et al., 2017*). Videos and images in color Doppler mode and B-mode were obtained using the Vevo1100 imaging system at a frequency of 50 MHz. Nikon NIS-Elements AR analysis and ImageJ software were employed for data extraction. To evaluate AV valve function, the ratio of inflow and outflow area in the same frame was quantified (*Gunawan et al., 2020*).

## Calcium imaging

At 120 hpf, embryos were treated with 10 mM 2,3-butanedione monoxime (Sigma, B0753) and mounted in 1% low melting agarose. Time-lapse images were acquired using a Nikon Ti2 microscope at a rate of 50 frames per second. Data were analyzed using Nikon-NIS Elements AR analysis software.

## Intracellular action potential recording

Electrophysiology study was performed on adult zebrafish ventricles as previously described (*Zhang et al., 2013*). Briefly, hearts were mounted in a chamber containing Tyrode's solution: NaCl 150 mM, KCl 5.4 mM, $MgSO_4$ 1.5 mM, $NaH_2PO_4$ 0.4 mM, $CaCl_2$ 2 mM, glucose 10 mM, HEPES 10 mM, pH was adjusted to 7.4. Glass pipettes with tip resistance 30–40 MΩ were filled with 3 M KCl solution. Intracellular action potentials were recorded using an HEKA amplifier and pClamp10.3 software (Molecular Devices).

## Histology and HE staining

Adult hearts were dissected and fixed overnight at 4°C in 4% PFA, followed by three times PBS washes. Dehydration involved an ethanol gradient (70%, 80%, 95%, 100%, 100%, 30 min each), followed by three soaks in dimethylbenzene at 65°C, before embedding in paraffin. Sections of 5 μm thickness were prepared using the Leica RM2235 manual rotary microtome for hematoxylin and eosin (HE) staining.

## Injection of mRNA

The embryonic zebrafish cDNA library was used as a template to amplify the *nrg1* and *tcf3b* fragment, which was then subcloned into the pCS2 vector. The vector was linearized using Not I restriction endonuclease, and mRNA was transcribed in vitro using the mRNA transcription kit (Ambion, AM1340). 100 pg of purified mRNA was injected into one-cell stage embryos.

## Luciferase assay

The LCR (luciferase reporter) plasmid was generated by subcloning the 5' UTR of *nrg1* into the upstream region of renilla luciferase on the psiCheck2 plasmid. Following construction, 25 pg of the LCR plasmid was co-injected with either 100 pg of *tcf3b* mRNA or 1 ng of *tcf3b* MO into one-cell stage zebrafish embryos. At 48 hpf, 20 embryos were gathered into one group and fully lysed. Subsequently, firefly and renilla luciferase activities were sequentially measured using a microplate reader with the dual luciferase reporter gene assay kit (Yeasen, 11402ES60), according to the manufacturer's instructions. The experiment was independently replicated three times. The relative renilla luciferase activity, normalized by firefly luciferase activity, served as an indicator of *nrg1* expression level under the influence of *tcf3b* overexpression or reduction.

## Image processing and statistical analysis

Whole-mount in situ hybridization images were captured using a Leica M165FC stereomicroscope. Live imaging of zebrafish embryos involved mounting anesthetized embryos in 1% low melting agarose (Sangon Biotech, A600015) and manually orienting them for optimal visual access to the heart. Confocal images were obtained with a Nikon Ti2 confocal microscope. Fluorescence intensity and cell number counting were processed using Nikon NIS-Elements AR analysis and ImageJ software. Statistical analysis was performed using GraphPad Prism 8 software. No statistical methods were used to predetermine sample size. Unpaired two-tailed Student's t-tests were used to determine statistical significance. Data are presented as mean ± s.e.m., *$p<0.05$ was considered to be statistically significant.

## Acknowledgements

We thank Dr. Pengfei Xu for providing morpholinos. We also thank Dr. Jia Li for the support in generating the *id2b:eGFP* line. This work was supported by the National Natural Science Foundation of China (32170823, 92468104, 31871462), and the National Key R&D Program of China (2023YFA1800600).

# Additional information

## Funding

| Funder | Grant reference number | Author |
| --- | --- | --- |
| National Natural Science Foundation of China | 32170823 | Peidong Han |
| National Key Research and Development Program of China | 2023YFA1800600 | Peidong Han |
| National Natural Science Foundation of China | 92468104 | Peidong Han |
| National Natural Science Foundation of China | 31871462 | Peidong Han |

The funders had no role in study design, data collection and interpretation, or the decision to submit the work for publication.

## Author contributions

Shuo Chen, Conceptualization, Data curation, Software, Formal analysis, Validation, Investigation, Visualization, Methodology, Writing - original draft; Jinxiu Liang, Conceptualization, Resources, Validation, Investigation, Visualization, Methodology, Writing - original draft, Writing - review and editing; Jie Yin, Software, Formal analysis, Methodology, Writing - original draft; Weijia Zhang, Data curation, Validation, Methodology; Peijun Jiang, Data curation, Visualization, Methodology; Wenyuan Wang, Resources, Data curation; Xiaoying Chen, Ruilin Zhang, Resources, Methodology; Yuanhong Zhou, Peng Xia, Fan Yang, Resources; Ying Gu, Resources, Supervision; Peidong Han, Conceptualization, Resources, Data curation, Supervision, Funding acquisition, Validation, Writing - original draft, Project administration, Writing - review and editing

## Author ORCIDs

Shuo Chen  https://orcid.org/0009-0005-5058-028X
Jie Yin  https://orcid.org/0000-0002-1166-2443
Ruilin Zhang  https://orcid.org/0000-0002-6594-450X
Peidong Han  https://orcid.org/0000-0002-6717-4000

## Ethics

All animal procedures were approved by the Animal Care and Use Committee of the Zhejiang University School of Medicine (application no.29296).

Joint Public Review: https://doi.org/10.7554/eLife.101151.3.sa1
Author response https://doi.org/10.7554/eLife.101151.3.sa2

# Additional files

## Supplementary files

MDAR checklist

## Data availability

The authors declare that all data supporting the findings in the paper are available in the article and the supplementary files. RNA-seq data have been deposited in GEO under accession number GSE295737 and GSE295738.

The following datasets were generated:

| Author(s) | Year | Dataset title | Dataset URL | Database and Identifier |
|---|---|---|---|---|
| Chen S, Yin J, Han P | 2025 | Effect of depletion of id2b on gene expression during zebrafish heart development | https://www.ncbi.nlm.nih.gov/geo/query/acc.cgi?acc=GSE295737 | NCBI Gene Expression Omnibus, GSE295737 |
| Chen S, Yin J, Han P | 2025 | Transcriptome analysis identifies id2b as a blood flow sensitive gene | https://www.ncbi.nlm.nih.gov/geo/query/acc.cgi?acc=GSE295738 | NCBI Gene Expression Omnibus, GSE295738 |

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
