## [Editor Report · eLife Assessment]

This study presents a **valuable** finding that the biomechanical force of heart contractility is required for robust endocardial id2b expression, which in return promotes valve development and myocardial function through upregulation of Neuregulin 1. The data were collected and analyzed using **solid** methodology and can be used as a starting point for deeper mechanistic insights into the genetic programs regulating endocardial-myocardial crosstalk during heart development.

---

## [Referee Report · Joint Public Review]

Summary:

How mechanical forces transmitted by blood flow contribute to cardiac development remains incompletely understood. Using the unique advantages of the zebrafish model, Chen et al make the fundamental discovery that endocardial expression of the transcriptional repressor, Id2b, is maintained in endocardial cells by blood flow. Id1b zebrafish mutants fail to form the valve in the atrioventricular canal (AVC) and show reduced myocardial contractility that they suggest is due to impaired calcium transients. Id2b mutants are largely viable during the first 6 months of life until ~20% display cardiomyopathy characterized by visible edema, structural abnormalities, retrograde blood flow, and reduced systolic function and calcium handling. Mechanistically, the authors suggest that flow-mediated expression of Id2b leads to neuregulin 1 (nrg1) upregulation by physically interacting with and sequestering the Tcf3b transcriptional repressor from conserved tcf3b binding sites upstream of nrg1. Overall, this study advances our understanding of flow-mediated endocardial-myocardial crosstalk during heart development.

Strengths:

The strengths of the study are the significance of the biological question being addressed, use of the zebrafish model, data quality, and use of genetic tools. The text is generally well-written and easy to understand.

Weaknesses:

The main weakness that remains is the lack of rigor surrounding the molecular mechanism where the authors suggest that blood flow induces endocardial expression of Id2b, which binds to Tcf3b and sequesters it from binding the Nrg1 promoter to repress transcription. Although good faith efforts were made to bolster their model, the physical interaction between Id2b and Tcf3b is limited to overexpression of tagged proteins in HEK293 cells. Moreover, no mutagenesis was performed on the tcf3b binding sites identified in the nrg1 promoter to learn their importance in vivo.

---

## [Author Response]

The following is the authors’ response to the original reviews

**Public Reviews:**

**Reviewer #1 (Public review):**
Summary:Chen et al. identified the role of endocardial id2b expression in cardiac contraction and valve formation through pharmaceutical, genetic, electrophysiology, calcium imaging, and echocardiography analyses. CRISPR/Cas9 generated id2b mutants demonstrated defective AV valve formation, excitation-contraction coupling, reduced endocardial cell proliferation in AV valve, retrograde blood flow, and lethal effects.Strengths:Their methods, data and analyses broadly support their claims.Weaknesses:The molecular mechanism is somewhat preliminary.

We thank the reviewer for the positive assessment of our work. A detailed point-by-point response has been incorporated in the response to “Recommendations for the authors” section.

**Reviewer #2 (Public review):**
Summary:Biomechanical forces, such as blood flow, are crucial for organ formation, including heart development. This study by Shuo Chen et al. aims to understand how cardiac cells respond to these forces. They used zebrafish as a model organism due to its unique strengths, such as the ability to survive without heartbeats, and conducted transcriptomic analysis on hearts with impaired contractility. They thereby identified id2b as a gene regulated by blood flow and is crucial for proper heart development, in particular, for the regulation of myocardial contractility and valve formation. Using both in situ hybridization and transgenic fish they showed that id2b is specifically expressed in the endocardium, and its expression is affected by both pharmacological and genetic perturbations of contraction. They further generated a null mutant of id2b to show that loss of id2b results in heart malformation and early lethality in zebrafish. Atrioventricular (AV) and excitation-contraction coupling were also impaired in id2b mutants. Mechanistically, they demonstrate that Id2b interacts with the transcription factor Tcf3b to restrict its activity. When id2b is deleted, the repressor activity of Tcf3b is enhanced, leading to suppression of the expression of nrg1 (neuregulin 1), a key factor for heart development. Importantly, injecting tcf3b morpholino into id2b-/- embryos partially restores the reduced heart rate. Moreover, treatment of zebrafish embryos with the Erbb2 inhibitor AG1478 results in decreased heart rate, in line with a model in which Id2b modulates heart development via the Nrg1/Erbb2 axis. The research identifies id2b as a biomechanical signaling-sensitive gene in endocardial cells that mediates communication between the endocardium and myocardium, which is essential for heart morphogenesis and function.Strengths:The study provides novel insights into the molecular mechanisms by which biomechanical forces influence heart development and highlights the importance of id2b in this process.Weaknesses:The claims are in general well supported by experimental evidence, but the following aspects may benefit from further investigation:(1) In Figure 1C, the heatmap demonstrates the up-regulated and down-regulated genes upon tricane-induced cardiac arrest. Aside from the down-regulation of id2b expression, it was also evident that id2a expression was up-regulated. As a predicted paralog of id2b, it would be interesting to see whether the up-regulation of id2a in response to tricane treatment was a compensatory response to the down-regulation of id2b expression.

We thank the reviewer for the comment. As suggested, we performed qRT-PCR analysis to assess *id2a* expression in tricaine-treated heart. Our results demonstrate a significant upregulation of *id2a* following the inhibition of cardiac contraction, suggesting a potential compensatory response to the decreased *id2b*. These new results have been incorporated into the revised manuscript (Figure 1D).

(2) The study mentioned that id2b is tightly regulated by the flow-sensitive primary cilia-klf2 signaling axis; however aside from showing the reduced expression of id2b in klf2a and klf2b mutants, there was no further evidence to solidify the functional link between id2b and klf2. It would therefore be ideal, in the present study, to demonstrate how Klf2, which is a transcriptional regulator, transduces biomechanical stimuli to Id2b.

We have examined the expression levels of *id2b* in both *klf2a* and *klf2b* mutants. The whole mount in situ results clearly demonstrate a decrease in *id2b* signal in both mutants (Figure 3E). As noted by the reviewer, *klf2* is a transcriptional regulator, suggesting that the regulation of *id2b* may occur at the transcriptional level. However, dissecting the molecular mechanisms underlying the crosstalk between *klf2* and *id2b* is beyond the scope of the present study.

(3) The authors showed the physical interaction between ectopically expressed FLAG-Id2b and HA-Tcf3b in HEK293T cells. Although the constructs being expressed are of zebrafish origin, it would be nice to show in vivo that the two proteins interact.

We thank the reviewer for this insightful comment. As suggested, we synthesized Flag-*id2b* and HA-*tcf3b* mRNA and co-injected them into 1-cell stage zebrafish embryos. We collected 100-300 embryos at 12, 24, and 48 hpf and performed western blot analysis using the same anti-HA and anti-Flag antibodies validated in HEK293 cell experiments. Despite multiple independent attempts, we were unable to detect clear bands of the tagged proteins in zebrafish embryos. We speculate that this could be due to mRNA instability, translational efficiency, or the low abundance of Id2b and Tcf3b proteins. We have acknowledged these technical limitations in the revised manuscript and clarified that the HEK293 cell data support a potential interaction between Id2b and Tcf3b, while confirming their endogenous interaction will require further investigations (Lines 295-296).

**Reviewer #3 (Public review):**
Summary:How mechanical forces transmitted by blood flow contribute to normal cardiac development remains incompletely understood. Using the unique advantages of the zebrafish model system, Chen et al make the fundamental discovery that endocardial expression of id2b is induced by blood flow and required for normal atrioventricular canal (AVC) valve development and myocardial contractility by regulating calcium dynamics. Mechanistically, the authors suggest that Id2b binds to Tcf3b in endocardial cells, which relieves Tcf3b-mediated transcriptional repression of Neuregulin 1 (NRG1). Nrg1 then induces expression of the L-type calcium channel component LRRC1. This study significantly advances our understanding of flow-mediated valve formation and myocardial function.Strengths:Strengths of the study are the significance of the question being addressed, use of the zebrafish model, and data quality (mostly very nice imaging). The text is also well-written and easy to understand.Weaknesses:Weaknesses include a lack of rigor for key experimental approaches, which led to skepticism surrounding the main findings. Specific issues were the use of morpholinos instead of genetic mutants for the bmp ligands, cilia gene ift88, and tcf3b, lack of an explicit model surrounding BMP versus blood flow induced endocardial id2b expression, use of bar graphs without dots, the artificial nature of assessing the physical interaction of Tcf3b and Id2b in transfected HEK293 cells, and artificial nature of examining the function of the tcf3b binding sites upstream of nrg1.

We thank the reviewer for the positive assessment and the constructive suggestions. We have performed additional experiments and data analysis to address these issues. A detailed point-by-point response has been incorporated in the response to “Recommendations for the authors” section.

**Recommendations for the authors:**

**Reviewer #1 (Recommendations for the authors):**
Questions/Concerns:(1) In the introduction, it would be beneficial to include background information on the id2b gene, what is currently known about its function in heart development/regeneration and in other animal models than just the zebrafish.

We thank the reviewer for the constructive suggestion. In the revised manuscript, we have added a paragraph in the Introduction to provide background on *id2b* and its role in heart development. Specifically, we discuss its function as a member of the ID (inhibitor of DNA binding) family of helix-loop-helix (HLH) transcriptional regulators and highlight its involvement in cardiogenesis in both zebrafish and mouse models. These additions help place our findings in a broader developmental and evolutionary context (Lines 91-100).

(2) Of the 6 differentially expressed genes identified in Figure 1C, why did the authors choose to focus on id2b and not the other 5 downregulated genes?

We thank the reviewer for the comments. As suggested, we have added a sentence in the revised manuscript to clarify the rationale for selecting *id2b* as the focus of the present study (Lines 117-121).

(3) As the authors showed representative in situ images for id2b expression with blebbistatin treatment in Figure 1E, and tnn2a MO in Figure 1F, it would also be beneficial to show relative mRNA expression levels for id2b in conditions of blebbistatin treatment and tnn2a MO knockdown. In Fig. 1C: id2b is downregulated with tricaine, but id2a is upregulated with tricaine. Do these genes perform similar or different functions, results of gene duplication events?

We thank the reviewer for the thoughtful suggestion. Our in situ hybridization results demonstrate reduced *id2b* expression following tricaine, blebbistatin, and *tnn2* morpholino treatment. To further validate these observations and enhance cellular resolution, we generated an *id2b:eGFP* knockin line. Analysis of this reporter line confirmed a significant reduction in *id2b* expression in the endocardium upon inhibition of cardiac contraction and blood flow (Figure 3A-D), supporting our in situ results. The divergent expression patterns of *id2a* and *id2b* in response to tricaine treatment likely reflect functional specification following gene duplication in zebrafish. While our current study focuses on characterizing the role of *id2b* in zebrafish heart development, the specific function of *id2a* remains to be determined.

(4) In Fig. 2b, could the authors compare the id2b fluorescence with RNAscope ISH at 24, 48, and 72 hpf? RNAscope ISH allows for the visualization of single RNA molecules in individual cells. The authors should at least compare these in the heart to demonstrate that id2b accurately reflects the endogenous id2b expression. In Fig. 2E: Suggest showing the individual fluorescent images for id2b:eGFP and kdrl:mCherry in the same colors as top panel images instead of in black and white. In Fig. 2F: The GFP fluorescence from id2b:eGFP signals looks overexposed.

We thank the reviewer for the valuable comment. In response, we attempted RNAscope in situ hybridization on embryos carrying the *id2b:eGFP* reporter to directly compare fluorescent reporter expression with endogenous *id2b* transcripts. However, we encountered a significant reduction in *id2b:eGFP* fluorescence following the RNAscope procedure, and even subsequent immunostaining with anti-GFP antibodies yielded only weak signals. Despite this technical limitation, the RNAscope results independently confirmed *id2b* expression in endocardial cells (Figure 2E), supporting the specificity and cell-type localization observed with the reporter line. As suggested by the reviewer, we have updated Figure 2G to display *id2b:eGFP* and *kdrl:mCherry* images in the same color scheme as the top panel to improve consistency and clarity. Additionally, we have replaced the images in Figure 2F to avoid overexposure and better represent the spatial distribution of *id2b:eGFP* in adult heart.

(5) In Fig. 3A: are all the images in panel A taken with the same magnification? In Fig. 3e, could the authors show the localization of klf2 and id2b and confirm their expression in the same endocardial cells? In Fig. 3, the authors conclude that klf2-mediated biomechanical signaling is essential for activating id2b expression. This statement is somewhat overstated because they only demonstrated that knockout of klf2 reduced id2b expression.

We thank the reviewer for these constructive comments. All images presented in Figure 3A were captured using the same magnification, as now clarified in the revised figure legend. We appreciate the reviewer’s question regarding the localization of *klf2* and *id2b*. While we were unable to directly visualize both markers in the same embryos due to the current unavailability of *klf2* reporter lines, prior studies using *klf2a:H2B-eGFP* transgenic zebrafish have demonstrated that *klf2a* is broadly expressed in endocardial cells, with enhanced expression in the atrioventricular canal region (Heckel et al., Curr Bio 2015, PMID: 25959969; Gálvez-Santisteban et al., Elife 2019, PMID: 31237233IF: NA NA B1). Our reporter analysis revealed a similarly broad endocardial expression pattern. These independent observations support the likelihood that and *id2b* are co-expressed in the same endocardial cell population.

We also appreciate the reviewer’s comments regarding the connection between biomechanical signaling and *id2b* expression. Previous studies have already established that biomechanical cues directly regulate *klf2* expression in zebrafish endocardial cells (Vermot et al., Plos Biol 2009, PMID: 19924233; Heckel et al., Curr Bio 2015, PMID: 25959969). In the present study, we observed a significant reduction in expression in both and *klf2b* mutants, suggesting that *id2b* acts downstream of *klf2*. These observations together establish the role of biomechanical cues-*klf2*-*id2b* signaling axis in endocardial cells. Nevertheless, we agree with the reviewer that further investigation is required to elucidate the precise mechanism by which *klf2* regulates *id2b* expression.

(6) In Fig. 4: What's the mRNA expression for id2b in WT and id2b mutant fish hearts?

We performed qRT-PCR analysis on purified zebrafish hearts and observed a significant reduction in *id2b* mRNA levels in *id2b* mutants compared to wild-type controls. These new results have been incorporated into the revised manuscript (Figure 4A).

(7) In Fig. 5E, the heart rate shows no difference between id2b+/+ and id2b-/- fish according to echocardiography analysis. However, Fig. 5B indicates a difference in heart rate. Could the authors explain this discrepancy?

We thank the reviewer for this insightful observation. In our study, we observed a reduction in heart rate in *id2b* mutants during embryonic stages (120 hpf), as shown in Figure 5B. However, this difference was not evident in adult fish based on echocardiography analysis (Figure 5E). While the exact reason for these changes during development remains unclear, it is possible that the reduction in cardiac output observed in *id2b* mutants during early development triggers compensatory mechanisms over time, ultimately restoring heart rate in adulthood. Given that heart rate is primarily regulated by pacemaker activity, further investigation will be required to determine whether such compensatory adaptations occur and to elucidate the underlying mechanisms.

(8) In Fig. 6A: it's a little hard to read the gene names in the left most image in the panel. In Fig. 6B, the authors conducted qRT-PCR analysis of 72 hpf embryonic hearts and validated decreased nrg1 levels in id2b-/- compared to control. Since nrg1 is not specifically expressed in endocardial cells in the developing heart, the authors should isolate endocardial cells and compare nrg1 expression in id2b-/- to control. This would ensure that the loss of id2b affects nrg1 expression derived from endocardial cells rather than other cell types. In Supp Figure S6: Suggest adding an image of the UMAP projection to show tcf3b expression in endocardial cells from sequencing analysis.

We thank the reviewer for these helpful suggestions. In response, we have increased the font size of gene names in the leftmost panel of Figure 6A to improve readability. Regarding *nrg1* expression, we acknowledge the importance of assessing its cell-type specificity. Unfortunately, due to the lack of reliable transgenic or knock-in tools for *nrg1*, its precise expression pattern in embryonic hearts remains unclear. We attempted to isolate endocardial cells from embryonic hearts using FACS, but the limited number of cells obtained at this stage precluded reliable qRT-PCR analysis. Nonetheless, our data show that *id2b* is specifically expressed in endocardial cells, and publicly available single-cell RNA-seq datasets also support that *nrg1* is predominantly expressed in endocardial, but not myocardial or epicardial cells during embryonic heart development (Figure 6-figure supplement 1). These findings suggest that *id2b* may regulate *nrg1* expression in a cell-autonomous manner within the endocardium. As suggested, we have also added a UMAP image to Figure 7-figure supplement 1 to show *tcf3b* expression in endocardial cells, further supporting the cell identity in single-cell dataset.

(9) In Fig. 6, Nrg1 knockout shows no gross morphological defects and normal trabeculation in larvae. Could the authors explain why they propose that endocardial id2b promotes nrg1 synthesis, thereby enhancing cardiomyocyte contractile function? Did Nrg1 knockdown with Mo lead to compromised calcium signaling and cardiac contractile function? Nrg2a has been reported to be expressed in endocardial cells in larvae, and its loss leads to heart function defects. Perhaps Nrg2a plays a more important role than Nrg1.

We thank the reviewer for raising this important point. Although we did not directly test *nrg1* knockout in our study, previous reports have shown that genetic deletion of *nrg1* in zebrafish does not impair cardiac trabeculation during embryonic stages (Rasouli et al., Nat Commun 2017, PMID: 28485381; Brown et al., J Cell Mol Med 2018, PMID: 29265764). However, reduced trabecular area and signs of arrhythmia were observed in juvenile and adult fish (Brown et al., J Cell Mol Med 2018, PMID: 29265764), suggesting a potential role for in maintaining cardiac structure and function later in development. Whether calcium signaling and cardiac contractility are affected at these stages remains to be determined. Given that morpholino-induced knockdown is limited to early embryonic stages, it is not suitable for assessing function in juvenile or adult hearts.

As noted by the reviewer, *nrg2a* is expressed in endocardial cells, and its deletion has been associated with cardiac defects (Rasouli et al., Nat Commun 2017, PMID: 28485381). To assess its potential involvement in our model, we performed qRT-PCR analysis and observed increased expression in *id2b* mutant hearts (Author response image 1). This upregulation may reflect a compensatory response to the loss of *id2b*. Therefore, *nrg2a* is unlikely to play an essential role in mediating the depressed cardiac function in this context.

**Author response image 1. sa2fig1:** Expression levels of *nrg2a*. qRT-PCR analysis of *nrg2a* mRNA in *id2b+/+* and *id2b-/-* adult hearts. Data were normalized to the expression of *actb1*. N=5 biological replicates, with each sample containing two adult hearts.

(10) In Fig. 7A of the IP experiment, it is recommended that the authors establish a negative control using control IgG corresponding to the primary antibody source. This control helps to differentiate non-specific background signal from specific antibody signal.

As suggested, we have included an IgG control corresponding to the primary antibody species in the immunoprecipitation (IP) experiment to distinguish specific from non-specific binding. The updated data are presented in Figure 7A of the revised manuscript.

(11) In Pg. 5, line 115: there is no reference included for previous literature on blebbistatin.

We have added the corresponding reference (Line 126, Reference #5).

In Pg. 5, lines 118-119; pg. 6 line 144: It would be beneficial to include a short sentence describing why choosing a tnnt2a morpholino knockdown to help provide mechanistic context to readers.

We thank the reviewer for the constructive suggestion. In cardiomyocytes, *tnnt2a* encodes a sarcomeric protein essential for cardiac contraction, and its knockdown is a well-established method for abolishing heartbeat and blood flow in zebrafish embryos, thereby allowing investigation of flow-dependent gene regulation. In the revised manuscript, we have added a sentence and corresponding reference to clarify the rationale for using *tnnt2a* morpholino in our study (Lines 128-129, Reference #35).

In Pg. 6, line 140: Results title of "Cardiac contraction promotes endocardial id2b expression through primary cilia but not BMP" is misleading and contradicts the results presented in this section and corresponding figure. For example, the bmp Mo knockdown experiments led to decreased id2b fluorescence and the last statement of this results section contradicts the title that BMP does not promote endocardial id2b in lines 179-180: "Collectively, these results suggest that BMP signaling and blood flow modulate id2b expression in a developmental-stage-dependent manner." It would be helpful to clarify whether BMP signaling is involved in id2b expression or not.

We apologize for any confusion caused by the section title. Our results demonstrate that *id2b* expression is regulated by both BMP signaling and biomechanical forces in a developmental-stage-specific manner. Specifically, morpholino-mediated knockdown of *bmp2b*, *bmp4*, and *bmp7a* at the 1-cell stage significantly reduced *id2b:eGFP* fluorescence at 24 hpf (Figure 3-figure supplement 1A, B), suggesting that *id2b* is responsive to BMP signaling during early embryonic development. However, treatment with the BMP inhibitor Dorsomorphin during later stages (24-48 or 36-60 hpf) did not significantly alter *id2b:eGFP* fluorescence intensity in individual endocardial cells, although a modest reduction in total endocardial cell number was noted (Figure 3-figure supplement 1C, D). These results suggest that BMP signaling is required for *id2b* expression during early development but becomes dispensable at later stages, when biomechanical cues may play a more prominent role. To address this concern and better reflect the data, we have revised the Results section title to: "BMP signaling and cardiac contraction regulate *id2b* expression". This revised title more accurately reflects the dual regulation of *id2b* expression (Line 153).

In line 205: Any speculation on why the hemodynamics was preserved between id2b mutant and WT siblings at 96 hpf?

As suggested, we have included a sentence to address this observation. “Surprisingly, the pattern of hemodynamics was largely preserved in *id2b-/-* embryos compared to *id2b+/+* siblings at 96 hpf (Figure 4-figure supplement 1E, Video 1, 2), suggesting that the reduced number of endocardial cells in the AVC region was not sufficient to induce functional defects.” (Lines 223-225)

In line 246: Fig. 6k and 6j are referenced, but should be figure 5k and 5j.

We have corrected this in the revised manuscript.

**Reviewer #2 (Recommendations for the authors):**
he manuscript was overall well explained, aside from a few minor points that would help facilitate reader comprehension:(1) The last paragraph of the introduction could be a brief summary of the study.

We thank the reviewer for this constructive suggestion. As recommended, we have included a paragraph in the Introduction section summarizing our key findings to provide clearer context for the study (Lines 96-100).

(2) Lines 127-128: 'revealed a substantial recapitulation of the... of endogenous id2b expression' may need to be rephrased.

We thank the reviewer for the valuable suggestion. In the revised manuscript, we have changed the sentence to: “Comparison of *id2b:eGFP* fluorescence with in situ hybridization at 24, 48, and 72 hpf revealed that the reporter signal closely recapitulates the endogenous *id2b* expression pattern.” (Lines 137-139)

(3) Line 182: '... in a developmental-stage-dependent manner' sounds a bit ambiguous, may need to slightly elaborate/ clarify what this means.

We thank the reviewer for the helpful comment. To improve clarity, we have revised the statement to: “Collectively, these results suggest that *id2b* expression is regulated by both BMP and biomechanical signaling, with the relative contribution of each pathway varying across developmental stages.” (Lines 195-197)

**Reviewer #3 (Recommendations for the authors):**
(1) The conclusion that BMP signaling prior to 24 hpf is necessary for id2b expression is not fully supported by the data. How do the authors envision pre-linear heart tube BMP signaling impacting endocardial id2b expression during later chamber stages? Id2b reporter fluorescence can be clearly visualized in the linear heart tube in panel B from Figure 1. Does id2b expression initiate prior to contraction? Can the model be refined by showing when id2b endocardial reporter fluorescence is first observed, and whether this early/pre-contractile expression is dependent on BMP signaling?

We thank the reviewer for the important comment. As suggested, we performed morpholino-mediated knockdown of *bmp2b*, *bmp4*, and *bmp7a* at the 1-cell stage. Live imaging at 24 hpf showed significantly reduced *id2b:eGFP* fluorescence compared to controls (Figure 3-figure supplement 1A, B), suggesting that *id2b* is responsive to BMP signaling during early embryonic development. However, treatment with the BMP inhibitor Dorsomorphin during 24-48 or 36-60 hpf did not significantly impact *id2b:eGFP* fluorescence intensity in individual endocardial cells, although a reduction in endocardial cell number was observed (Figure 3-figure supplement 1C, D). These results suggest that BMP signaling is essential for *id2b* expression during early embryonic development, while it becomes dispensable at later stages, when biomechanical cues exert a more significant role.

(2) Overexpressing tagged versions of TCF3b and Id2b in HEK293 cells is a very artificial way to make the major claim that these two proteins interact in endogenous endocardial cells. Can this be done in zebrafish embryonic or adult hearts?

We thank the reviewer for this insightful comment. As suggested, we synthesized Flag-*id2b* and HA-*tcf3b* mRNA and co-injected them into 1-cell stage zebrafish embryos. We collected 100-300 embryos at 12, 24, and 48 hpf and performed western blot analysis using the same anti-HA and anti-Flag antibodies validated in HEK293 cell experiments. Despite multiple independent attempts, we were unable to detect clear bands of the tagged proteins in zebrafish embryos. We speculate that this could be due to mRNA instability, translational efficiency, or the low abundance of Id2b and Tcf3b proteins. We have acknowledged these technical limitations in the revised manuscript and clarified that the HEK293 cell data support a potential interaction between Id2b and Tcf3b, while confirming their endogenous interaction will require further investigations (Lines 295-296).

(3) The data presented are consistent with the claim that the tcf3b binding sites are functional upstream of nrg1 to repress its transcription. To fully support this idea, those two sites should be disrupted with gRNAs if possible.

We thank the reviewer for the valuable suggestion. In response, we attempted to disrupt the *tcf3b* binding sites using sgRNAs. However, we encountered technical difficulties in identifying sgRNAs that specifically and efficiently target these binding sites without affecting adjacent regions. Despite these challenges, our luciferase reporter assay, using *tcf3b* mRNA overexpression and morpholino knockdown, clearly demonstrated that *tcf3b* binds to and regulates *nrg1* promoter region. Nevertheless, we acknowledge that future study using genome editing will be necessary to validate the direct binding of *tcf3b* to *nrg1* promoter.

Minor Points:(1) Must remove all of the "data not shown" statements and add the primary data to the Supplemental Figures.

As suggested, we have removed all of the “data not shown” statements and added the original data to the revised manuscript (Figure 4E, middle panels, and Figure 4-figure supplement 1F)

(2) Must present the order of the panels in the figure as they are presented in the text. One example is Figure 6 where 6E is discussed in the text before 6C and 6D.

We thank the reviewer for bring up this important point. In the revised manuscript, we have carefully revised the manuscript to ensure that the order of figure panels matches the sequence in which they are discussed in the text. Specifically, we have reorganized the presentation of Figure 6 panels to align with the text flow, discussing panels 6C and 6D before panel 6E. The updated figure and corresponding text have been corrected accordingly in the revised manuscript.

(3) Change the italicized gene names (e.g. tcf3b) to non-italicized names with the first letter capitalized (e.g. Tcf3b) when referencing the protein.

As suggested, we have revised the manuscript to use non-italicized names with the first letter capitalized when referring to proteins.

(4) All bar graphs should be replaced with dot bar graphs.

We have replaced all bar graphs with dot bar graphs throughout the manuscript.

(5) The new id2b mutant allele should be validated as a true null using quantitative RT-PCR to show that the message becomes destabilized through non-sense mediated decay or by immunostaining/western blot analysis if there is a zebrafish Id2b-specific antibody available.

We thank the reviewer for this important suggestion. We have performed qRT-PCR analysis and detected a significant reduction in *id2b* mRNA levels in *id2b-/-* compared to *id2b+/+* controls. These new results are presented in Figure 4A of the revised manuscript.

(6) Was tricaine used to anesthetize embryos for capturing heart rate and percent fractional area change? This analysis should be performed with no or very limited tricaine as it affects heart rate and systolic function. These parameters were captured at 120 hpf, but the authors should also look earlier at 72 hpf at a time when valves are not present by calcium transients are necessary to support heart function.

We thank the reviewer for this important comment. When performing live imaging to assess cardiac contractile function, we used low-dose tricaine (0.16 mg/mL) to anesthetize the zebrafish embryos. We have included this important information in the Methods section (Line 503). As suggested, we have also included the heart function results at 72 hpf, which are now presented in Figure 5-figure supplement 2A-C of the revised manuscript.

(7) The alpha-actinin staining in Figure 5-supplement 2D is very pixelated and unconvincing. This should be repeated and imaged at a higher resolution.

As suggested, we have re-performed the α-actinin staining and acquired higher-resolution images. The updated results are now presented in Figure 5-figure supplement 2G of the revised manuscript.

(8) The authors claim that reductions in id2b mutant heart contractility are due to perturbed calcium transients instead of sarcomere integrity. Why do the authors think that regulation of calcium dynamics was not observed in the DEG enriched GO-terms? Was significant downregulation of cacna1 identified in the bulk RNAseq?

We thank the reviewer for raising this important point. In our bulk RNAseq dataset comparing *id2b* mutant and control hearts, GO term enrichment was primarily associated with pathways related to cardiac muscle contraction and heart contraction (Figure 5-figure supplement 1B). We speculate that the transcriptional changes related to calcium dynamics may be relatively subtle and thus were not captured as significantly enriched GO terms. In addition, our qRT-PCR analysis revealed a significant reduction in *cacna1c* expression in *id2b* mutant hearts compared to controls, suggesting that *id2b* deletion impairs calcium channel expression. However, this change was not detected by RNA-seq, likely due to limitations in sensitivity.

(9) In line 277, the authors say, "To determine whether this interaction occurs in zebrafish, Flag-id2b and HA-tcf3b were co-expressed in HEK293 cells...". This should be re-phrased to, "To determine if zebrafish Id2b and Tcf3b interact in vitro, Flag-id2b and HA-tcf3b were co-expressed in HEK293 cells for co-immunoprecipitation analysis." The sentence in line 275 should be changed to, "....heterodimer with Tcf3b to limit its function as a potent transcriptional repressor."

We thank the reviewer for these constructive comments and have revised the text accordingly (Lines 291-294).

(10) Small text corrections or ideas:Line 63: emphasized

We have corrected this in the revised manuscript.

Line 71: studied signaling pathways

We have corrected this in the revised manuscript.

Line 106: the top 6 DEGS (I think that the authors mean top 6 GO-terms) and is Id2b in one of the enriched GO categories?

*id2b* is one of the top DEGs. This point has been clarified in the revised manuscript (Lines 116-117).

Line 125: a knockin id2b:eGFP reporter line

We have corrected this in the revised manuscript (Line 136).

Line 138: This paragraph could use a conclusion sentence.

We have added a conclusion sentence in the revised manuscript (Lines 150-151).

Line 190: id2b-/- zebrafish experienced early lethality

We have revised the statement as suggested (Line 206).

Line 193: The prominent enlargement of the atrium with a smaller ventricle has characterized as cardiomyopathy in zebrafish (Weeks et al. Cardiovasc Res, 2024, PMID: 38900908), which has also been associated with disruptions in calcium transients (Kamel et al J Cardiovasc Dev Dis, 2021, PMID: 33924051 and Kamel et al, Nat Commun 2021, PMID: 34887420). This information should be included in the text along with these references.

We thank the reviewer for this helpful suggestion. We have incorporated these important references into the revised manuscript and included the relevant information to acknowledge the established link between atrial enlargement, cardiomyopathy, and disrupted calcium transients in zebrafish models (Reference #41, 42, and 45; Lines 210 and 260).